# Thermodynamics of structure-forming systems

Jan Korbel [1,2], Simon David Lindner[1,2], Rudolf Hanel[1,2] & Stefan Thurner [1,2,3 ✉]

Structure-forming systems are ubiquitous in nature, ranging from atoms building molecules to self-assembly of colloidal amphibolic particles. The understanding of the underlying thermodynamics of such systems remains an important problem. Here, we derive the entropy for structure-forming systems that differs from Boltzmann-Gibbs entropy by a term that explicitly captures clustered states. For large systems and low concentrations the approach is equivalent to the grand-canonical ensemble; for small systems we find significant deviations. We derive the detailed fluctuation theorem and Crooks' work fluctuation theorem for structure-forming systems. The connection to the theory of particle self-assembly is discussed. We apply the results to several physical systems. We present the phase diagram for patchy particles described by the Kern-Frenkel potential. We show that the Curie-Weiss model with molecule structures exhibits a first-order phase transition.

[1] Section for the Science of Complex Systems, CeMSIIS, Medical University of Vienna, Vienna, Austria. [2] Complexity Science Hub Vienna, Vienna, Austria. [3] Santa Fe Institute, Santa Fe, NM, USA. ✉email: stefan.thurner@meduniwien.ac.at

Ludwig Boltzmann defined entropy as the logarithm of state multiplicity. The multiplicity of independent (but possibly interacting) systems is typically given by multinomial factors that lead to the Boltzmann–Gibbs entropy and the exponential growth of phase space volume as a function of the degrees of freedom. In recent decades, much attention was given to systems with long-range and coevolving interactions that are sometimes referred to as complex systems[1]. Many complex systems do not exhibit an exponential growth of phase space[2–5]. For correlated systems, it typically grows subexponentially[6–14], systems with superexponential phase space growth were recently identified as those capable of forming structures from its components[5,15]. A typical example of this kind are complex networks[16], where complex behavior may lead to ensemble inequivalence[17]. The most prominent example of structure-forming systems are chemical reaction networks[18–20]. The usual approach to chemical reactions— where free particles may compose molecules—is via the grand-canonical ensemble, where particle reservoirs make sure that the number of particles is conserved on average. Much attention has been given to finite-size corrections of the chemical potential[21,22] and nonequilibrium thermodynamics of small chemical networks[23–26]. However, for small closed systems, fluctuations in particle reservoirs might become nonnegligible and predictions from the grand-canonical ensemble become inaccurate. In the context of nanotechnology and colloidal physics, the theory of self-assembly[27] gained recent interest. Examples of self-assembly include lipid bilayers and vesicles[28], microtubules, molecular motors[29], amphibolic particles[30], or RNA[31]. The thermodynamics of self-assembly systems has been studied, both experimentally and theoretically, often dealing with particular systems, such as Janus particles[32]. Theoretical and computational work have explored self-assembly under nonequilibrium conditions[33,34]. A review can be found in Arango-Restrepo et al.[35].

Here, we present a canonical approach for closed systems where particles interact and form structures. The main idea is to start not with a grand-canonical approach to structure-forming systems but to see within a canonical description which terms in the entropy emerge that play the role of the chemical potential in large systems. A simple example for a structure-forming system, the magnetic coin model, was recently introduced in Jensen et al.[15]. There $n$ coins are in two possible states (head and tail), and in addition, since coins are magnetic, they can form a third state, i.e., any two coins might create a bond state. The phase space of this model, $W(n)$, grows superexponentially, $W(n) \sim n^{n/2} e^{2\sqrt{n}} \sim e^{n\log n}$. We first generalize this model to arbitrary cluster sizes and to an arbitrary number of states. We then derive the entropy of the system from the corresponding log multiplicity and use it to compute thermodynamic quantities, such as the Helmholtz free energy. With respect to Boltzmann–Gibbs entropy, there appears an additional term that captures the molecule states. By using stochastic thermodynamics, we obtain the appropriate second law for structure-forming systems and derive the detailed fluctuation theorem. Under the assumption that external driving preserves microreversibility, i.e., detailed balance of transition rates in quasi-stationary states, we derive the nonequilibrium Crooks' fluctuation theorem for structure-forming systems. It relates the probability distribution of the stochastic work done on a nonequilibrium system to thermodynamic variables, such as the partial Helmholtz free energy, temperature, and size of the initial and final cluster states. Finally, we apply our results to several physical systems: we first calculate the phase diagram for the case of patchy particles described by the Kern–Frenkel potential. Second, we discuss the fully connected Ising model where molecule formation is allowed. We show that the usual second-order transition in the fully connected Ising model changes to first-order.

## Results

**Entropy of structure-forming systems**. To calculate the entropy of structure-forming systems, we first define a set of possible microstates and mesostates. Let us consider a system of $n$ particles. Each single particle can attain states from the set $\mathcal{X}^{(1)} = \{x_1^{(1)}, \ldots, x_{m_1}^{(1)}\}$. The superscript number (1) indicates that the states correspond to a single-particle state, and $m_1$ denotes the number of these states. A typical set of states could be the spin of the particle $\{\uparrow, \downarrow\}$, or a set of energy levels. Having only single-particle states, the microstate of the system consisting of $n$ particles is a vector $(X_1, X_2, \ldots, X_n)$, where $X_k \in \mathcal{X}^{(1)}$ is the state of $k$th particle. Let us now assume that any two particles can create a two-particle state. This two-particle state can be a molecule composed of two atoms, a cluster of two colloidal particles, etc. We call this state as a cluster. This two-particle cluster can attain states $\mathcal{X}^{(2)} = \{x_1^{(2)}, \ldots, x_{m_2}^{(2)}\}$. A microstate of a system of $n$ particles is again a vector $(X_1, X_2, \ldots, X_n)$, but now either $X_k \in \mathcal{X}^{(1)}$ or $X_k \in \mathcal{X}^{(2)} \times \mathbb{Z}_n^2$. For instance, a state of particle $k$ belonging to a two-particle cluster can be written as $X_k = x_1^{(2)}(k_1, k_2)$. The indices in the brackets tell us that the particle $k$ belongs to the cluster of size two in the state $x_1^{(2)}$ and the cluster is formed by particles $k_1$ and $k_2$ ($k_1 < k_2$). Indeed, either $k_1 = k$ or $k_2 = k$.

Now assume that particles can also form larger clusters up to a maximal size, $m$. Consider $m$ as a fixed number, $m \leq n$. Generally, clusters of size $j$ have states $\mathcal{X}^{(j)} = \{x_1^{(j)}, \ldots, x_{m_j}^{(j)}\}$. The corresponding states of the particle are always elements from sets $\mathcal{X}^{(j)} \times \mathbb{Z}_n^j$ with the restriction that if the $k$th particle is in a state $x_i^{(j)}(k_1, \ldots, k_j)$ then $k_l < k_{l+1}$, for all $l$ and one $k_l = k$. Consider an example of four particles. Particles are either in a free state or they form a cluster of size two. A state of each particle is either $s^{(1)}$—a free particle, or $x^{(2)}(i, j)$—a cluster compound from particles $i$ and $j$. As an example, a typical microstate is $\Psi = (x^{(1)}, x^{(2)}(2,3), x^{(2)}(2,3), x^{(1)})$, which means that particles 1 and 4 are free and particles 2 and 3 form a cluster.

Now consider a mesoscopic scale, where the mesostate of the system is given only by the number of clusters in each state $x_i^{(j)}$. Let us denote $n_i^{(j)}$ as the number of clusters in state $x_i^{(j)}$. The mesostate is therefore characterized by a vector $\mathbb{N} = \left(n_i^{(j)}\right)$, which corresponds to a frequency (histogram) of microstates. The normalization condition is given by the fact that the total number of particles is $n$, i.e., $\sum_{ij} j n_i^{(j)} = n$. For example, a mesostate, $\mathbb{N}_\Psi$, corresponding to a microstate $\Psi$ is $\mathbb{N}_\psi = \left(n^{(1)} = 2, n^{(2)} = 1\right)$, denoting that there are two free particles and one two-particle cluster.

The Boltzmann entropy[36] of this mesostate is given by

$$S(\mathbb{N}) = \log W(\mathbb{N}), \tag{1}$$

where $W$ is the multiplicity of the mesostate, which is the number of all distinct microstates corresponding to the same mesostate. To determine the number of all distinct microstates corresponding to a given mesostate, let us order the particles and number them from 1 to $n$. By permutation of the particles we obtain the different possible microstates. The number of all permutations is simply $n!$. However, some permutations correspond to the same microstate and we are overcounting. In our example with one cluster and two free particles, the permutations (4, 2, 3, 1) and (1, 3, 2, 4) correspond to the same microstate $\Psi = (x^{(1)}, x^{(2)}(2, 3), x^{(2)}(2, 3), x^{(1)})$. However, permutation (2, 1, 3, 4) corresponds to the microstate $\Psi' = (x^{(2)}(1, 3), x^{(1)}, x^{(2)}(1, 3), x^{(1)})$. This microstate

is a distinct microstate corresponding to the same mesostate, $\mathbb{N}_\Psi \equiv \mathbb{N}_{\Psi'} = (n^{(1)} = 2, n^{(2)} = 1)$.

The number of microstates giving the same mesostate can be expressed as the product of configurations with the same state for each $x_i^{(j)}$. Let us begin with the particles that do not form clusters. The number of equivalent representations for one distinct state is $\left(n_i^{(1)}\right)!$, which corresponds to the number of permutations of all particles in the same state. For the cluster states, one can think about equivalent representations of one microstate in two steps: first permute all clusters, which gives $\left(n_i^{(j)}\right)!$ possibilities. Then, permute the particles in the cluster, which gives $j!$ possibilities for every cluster, so that we end up with $(j!)^{n_i^{(j)}}$ combinations.

As an example, consider the case of four particles. First, we look at free particles that attain states $x_1^{(1)}$ or $x_2^{(1)}$. Let us consider a mesostate $\mathbb{N}_1 = \left(n_1^{(1)} = 2, n_2^{(1)} = 2\right)$, i.e., two particles in the first state and two particles in the second. The number of distinct microstates corresponding to the mesostate $\mathbb{N}_1$ is given by $W(\mathbb{N}_1) = 4!/(2!2!) = 6$. All microstates that belong to the mesostate $\mathbb{N}_1$ are

$$(x_1^{(1)}, x_1^{(1)}, x_2^{(1)}, x_2^{(1)}) \quad (x_1^{(1)}, x_2^{(1)}, x_1^{(1)}, x_2^{(1)})$$
$$(x_1^{(1)}, x_2^{(1)}, x_2^{(1)}, x_1^{(1)}) \quad (x_2^{(1)}, x_2^{(1)}, x_1^{(1)}, x_1^{(1)})$$
$$(x_2^{(1)}, x_1^{(1)}, x_2^{(1)}, x_1^{(1)}) \quad (x_2^{(1)}, x_1^{(1)}, x_1^{(1)}, x_2^{(1)})$$

Now imagine that the four particles are either free or form two-particle clusters. The microstate of a particle is either $x^{(1)}$ or $x^{(2)}(i,j)$. Let us consider a mesostate $\mathbb{N}_2 = \left(n^{(1)} = 0, n^{(2)} = 2\right)$, i.e., two clusters of size two. The number of distinct microstates is just $W(\mathbb{N}_2) = 4!/(2!(2!)^2) = 3$. The microstates corresponding to the mesostate $\mathbb{N}_2$ are

$$(x^{(2)}(1,2), x^{(2)}(1,2), x^{(2)}(3,4), x^{(2)}(3,4))$$
$$(x^{(2)}(1,3), x^{(2)}(2,4), x^{(2)}(1,3), x^{(2)}(2,4))$$
$$(x^{(2)}(1,4), x^{(2)}(2,3), x^{(2)}(2,3), x^{(2)}(1,4))$$

For example, a microstate $(x^{(2)}(2,1), x^{(2)}(2,1), x^{(2)}(4,3), x^{(2)}(4,3))$ is the same as the first microstate because we just relabel $1\leftrightarrow2$ and $3\leftrightarrow4$. In summary, the multiplicity corresponding to $x_i^{(j)}$ is $(n_i^{(j)})!(j!)^{n_i^{(j)}}$, and we can express the total multiplicity as

$$W(\mathbb{N}) = \frac{n!}{\prod_{ij}\left(\left(n_i^{(j)}\right)!(j!)^{n_i^{(j)}}\right)}. \tag{2}$$

Using Stirling's formula $\log n! \approx n\log n - n$, we get for the entropy

$$S(\mathbb{N}) \approx n\log n - n$$
$$- \sum_{ij}\left(n_i^{(j)}\log n_i^{(j)} - n_i^{(j)} + n_i^{(j)}\log j!\right). \tag{3}$$

Using the normalization condition, $n = \sum_{ij} j n_i^{(j)}$, and combining the first term with the remaining ones, we get the entropy per particle in terms of ratios $\wp_i^{(j)} = n_i^{(j)}/n$

$$\mathcal{S}(\mathbb{N}) = \frac{S(\{n_i^{(j)}\})}{n} = -\sum_{ij}\left[\frac{n_i^{(j)}}{n}\log\left(\frac{n_i^{(j)}}{n}\right)\right.$$
$$\left. -\frac{n_i^{(j)}}{n}\log\left(\frac{j!}{n^{j-1}}\right) - \frac{n_i^{(j)}}{n} + \frac{j n_i^{(j)}}{n}\right]. \tag{4}$$

Normalization is given by $\sum_{ij} j\wp_i^{(j)} = 1$. Therefore, $p_i^{(j)} = j\wp_i^{(j)}$ can be interpreted as the probability that a particle is a part of a cluster

in state $x_i^{(j)}$. On the other hand, the quantity $\wp_i^{(j)}$ is the relative number of clusters. Since $\sum_{ij}\frac{j n_i^{(j)}}{n} = 1$, we neglect the constant without changing the thermodynamic relations.

In the remainder, we denote thermodynamic quantities per particle by calligraphic script and total quantities by normal script. We express the entropy per particle as

$$\mathcal{S}(\wp) = -\sum_{ij}\wp_i^{(j)}\left(\log\wp_i^{(j)} - 1\right)$$
$$-\sum_{ij}\wp_i^{(j)}\log\left(\frac{j!}{n^{j-1}}\right), \tag{5}$$

or equivalently in terms of the probability distribution, $p_i^{(j)}$, as

$$\mathcal{S}(P) = -\sum_{ij}\frac{p_i^{(j)}}{j}\left(\log\frac{p_i^{(j)}}{j} - 1\right)$$
$$-\sum_{ij}\frac{p_i^{(j)}}{j}\log\left(\frac{j!}{n^{j-1}}\right). \tag{6}$$

*Finite interaction range.* Up to now, we assumed an infinite range of interaction between particles, which is unrealistic for chemical reactions, where only atoms within a short range form clusters. A simple correction is obtained by dividing the system into a fixed number of boxes: particles within the same box can form clusters, particles in different boxes cannot. We begin by calculating the multiplicity for two boxes. For simplicity, assume that they both contain $n/2$ particles. The multiplicity of a system with two boxes, $\tilde{W}\left(n_i^{(j)}\right)$, is given by the sum of all possible partitions of $n_i^{(j)}$ clusters with state $x_i^{(j)}$ into the first box (containing $^1 n_i^{(j)}$ clusters) and the second box (containing $^2 n_i^{(j)}$ clusters), such that $n_i^{(j)} = {}^1 n_i^{(j)} + {}^2 n_i^{(j)}$. The multiplicity is therefore

$$\tilde{W}\left(n_i^{(j)}\right) = \sum_{^1 n_i^{(j)} + {}^2 n_i^{(j)} = n_i^{(j)}} W\left(^1 n_i^{(j)}\right) W\left(^2 n_i^{(j)}\right), \tag{7}$$

where $W$ is the multiplicity in Eq. (2). The dominant contribution to the sum comes from the term, where $^1 n_i^{(j)} = {}^2 n_i^{(j)} = n_i^{(j)}/2$, so that we can approximate the multiplicity by $\tilde{W}(n_i^{(j)}) \approx W(n_i^{(j)}/2)^2$. Similarly, for $b$ boxes we obtain the multiplicity

$$\tilde{W}(n_i^{(j)}) = W(n_i^{(j)}/b)^b = \frac{[(n/b)!]^b}{\prod_{ij}\left(\left[(n_i^{(j)}/b)!\right]^b (j!)^{n_i^{(j)}}\right)}. \tag{8}$$

By defining the concentration of particles as $c = n/b$, the entropy per particle becomes

$$\mathcal{S}(\wp) = -\sum_{ij}\wp_i^{(j)}\left(\log\wp_i^{(j)} - 1\right)$$
$$-\sum_{ij}\wp_i^{(j)}\log\left(\frac{j!}{c^{j-1}}\right), \tag{9}$$

or, respectively,

$$\mathcal{S}(P) = -\sum_{ij}\frac{p_i^{(j)}}{j}\left(\log\frac{p_i^{(j)}}{j} - 1\right)$$
$$-\sum_{ij}\frac{p_i^{(j)}}{j}\log\left(\frac{j!}{c^{j-1}}\right). \tag{10}$$

Note that the entropy of structure-forming systems is both additive and extensive in the sense of Lieb and Yngvason[37]. It is

also concave, ensuring the uniqueness of the maximum entropy principle. For more details and connections to axiomatic frameworks, see Supplementary Discussion.

**Equilibrium thermodynamics of structure-forming systems.** We now focus on the equilibrium thermodynamics obtained, for example, by considering the maximum entropy principle. Consider the internal energy

$$U(n_i^{(j)}) = \sum_{ij} n_i^{(j)} \epsilon_i^{(j)} = n \sum_{ij} \wp_i^{(j)} \epsilon_i^{(j)} = n\, \mathcal{U}(\wp_i^{(j)}). \quad (11)$$

Using Lagrange multipliers to maximize the functional

$$\mathcal{S}(\wp) - \alpha \left( \sum_{ij} j \wp_i^{(j)} - 1 \right) - \beta \left( \sum_{ij} \wp_i^{(j)} \epsilon_i^{(j)} - \mathcal{U} \right), \quad (12)$$

leads to the following:

$$-\log \hat{\wp}_i^{(j)} - \log \left( \frac{j!}{c^{j-1}} \right) - \alpha j - \beta \epsilon_i^{(j)} = 0, \quad (13)$$

and the resulting distribution is

$$\hat{\wp}_i^{(j)} = \frac{c^{j-1}}{j!} \exp\left( -j\alpha - \beta \epsilon_i^{(j)} \right). \quad (14)$$

Here, we introduce the partial partition functions, $\mathcal{Z}_j = \frac{c^{j-1}}{j!} \sum_i e^{-\beta \epsilon_i^{(j)}}$, and the quantity $\Lambda = e^{-\alpha}$. $\Lambda$ is obtained from

$$\sum_{ij} j \hat{\wp}_i^{(j)} = \sum_{j=1}^m j\, \mathcal{Z}_j\, \Lambda^j = 1, \quad (15)$$

which is a polynomial equation of order $m$ in $\Lambda$. The connection with thermodynamics follows through Eq. (13). By multiplying with $\hat{\wp}_i^{(j)}$ and summing over $i,j$, we get $\mathcal{S}(\wp) - \sum_{ij} \hat{\wp}_i^{(j)} - \alpha - \beta \mathcal{U} = 0$. Note that $\sum_{ij} \hat{\wp}_i^{(j)} = \sum_{ij} \hat{n}_i^{(j)}/n = M/n = \mathcal{M}$ is the number of clusters, divided by the number of particles in the system. The number of clusters per particle is

$$\mathcal{M} = \sum_{ij} \hat{\wp}_i^{(j)} = \sum_j \mathcal{Z}_j\, \Lambda^j. \quad (16)$$

The Helmholz free energy is thus obtained as

$$\mathcal{F} = \mathcal{U} - \frac{1}{\beta} \mathcal{S} = -\frac{\alpha}{\beta} - \frac{1}{\beta} \mathcal{M}. \quad (17)$$

Finally, we can write the total partition function as

$$\mathcal{Z} = \exp(-\beta \mathcal{F}) = \frac{1}{\Lambda} \prod_{j=1}^m \exp(\Lambda^j \mathcal{Z}_j). \quad (18)$$

*Comparison with the grand-canonical ensemble.* To compare the presented exact approach with the grand-canonical ensemble, consider the simple chemical reaction, $2X \rightleftharpoons X_2$. Without loss of generality, assume that free particles carry some energy, $\epsilon$. We calculate the Helmholtz free energy for both approaches in Supplementary Information. In Fig. 1, we show the corresponding specific heat, $c(T) = -T \frac{\partial^2 \mathcal{F}}{\partial T^2}$. For large systems, the usual grand-canonical ensemble approach and the exact calculation with a strictly conserved number of particles converge. For small systems, however, there appear notable differences. This is visible in Fig. 1, where only for large $n$ and low concentrations, $c$, the specific heat for the exact approach (squares) and the grand-canonical ensemble (triangles) become identical. The inset shows the ratio of the specific heat, $c_C/c_{GC} - 1$, vanishing for large $n$. For large systems, the exact approach and the the grand-canonical ensemble are equivalent.

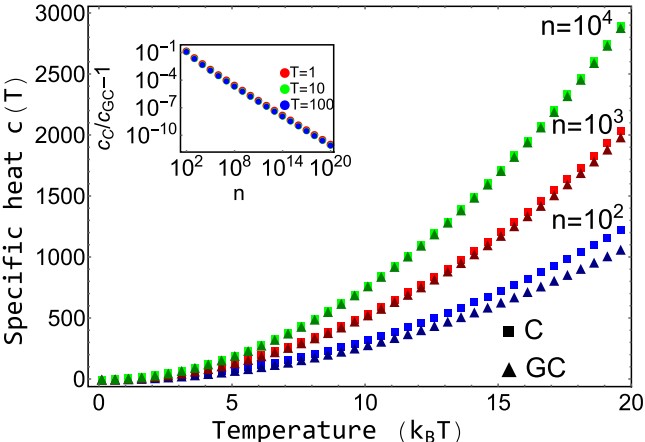

**Fig. 1 Specific heat, $c(T)$, for the reaction $2X \rightleftharpoons X_2$ for the presented canonical approach with an exact number of particles in comparison to the grand-canonical ensemble.** The specific heat for the canonical ensemble (C) is drawn by squares, and the specific heat for the grand-canonical ensemble (GC) is drawn by triangles. $n$ denotes the number of particles. For small systems the difference of the approaches becomes apparent. The inset shows the ratio of the specific heat calculated from the exact approach to the one obtained from the grand-canonical ensemble, $c_C/c_{GC} - 1$. For large $n$ the quantity decays to zero for any temperature.

**Relation to the theory of self-assembly.** In many applications, the number of energetic configurations for each cluster size is so large that one is only interested in the distribution of cluster sizes. For this case, it is possible to formulate an effective theory considering contributions from all configurations that is known as the theory of self-assembly. For an overview, see Likos et al.[27].

To compute the free energy in terms of the cluster-size distribution, we define the latter as

$$\hat{\wp}^{(j)} = \sum_i \hat{\wp}_i^{(j)} = \Lambda^j \mathcal{Z}_j. \quad (19)$$

This is the distribution obtained from a free energy of the ideal gas of clusters, as discussed in Fantoni et al.[32] for the case of Janus particles and in Vissersa et al.[38] for the more general case of one-patch colloids. The entropy of the relative cluster size can be introduced as

$$\mathcal{S}_c(\wp) = -\sum_{j=1}^m \wp^{(j)} \left( \log \wp^{(j)} - 1 \right). \quad (20)$$

By introducing the partial free energy as

$$\Phi_j = -\frac{1}{\beta} \log \mathcal{Z}_j, \quad (21)$$

the energy constraint takes the form of the expected free energy, averaged over cluster size, $\Phi = \sum_{j=1}^m \wp^{(j)} \Phi_j$. The cluster-size distribution is obtained by maximization of the functional

$$\mathcal{S}_c(\wp) - \alpha_c \left( \sum_{j=1}^m j \wp^{(j)} - 1 \right) - \beta \left( \sum_{j=1}^m \wp^{(j)} \Phi_j - \Phi \right). \quad (22)$$

It is clear that Eq. (19) is the solution of the maximization. The free energy can be now expressed as

$$\mathcal{F}_c = \Phi - \frac{1}{\beta} \mathcal{S}_c = -\frac{\alpha_c}{\beta} - \frac{\mathcal{M}}{\beta}, \quad (23)$$

which has the same structure as when calculated in terms of $\wp_i^{(j)}$.

**Examples for thermodynamics of structure-forming systems.** We now apply the results obtained in the previous section to several examples of structure-forming systems. We particularly focus on how the presence of mescoscopic structures of clustered states leads to the macroscopic physical properties. In the presence of structure formation, there exists a phase transition between a free particle fluid phase and a condensed phase, containing clusters of particles. This phase transition is demonstrated in two examples.

The first example on soft-matter self-assembly describes the process of condensation of one-patch colloidal amphibolic particles. This condensation is relevant in applications in nanomaterials and biophysics. The second example covers the phase transition of the Curie–Weiss spin model for the situation where particles form molecules. In Supplementary Information, we discuss the additional examples of a magnetic gas and a size-dependent chemical potential.

*Kern–Frenkel model of patchy particles.* Recently, the theory of soft-matter self-assembly has successfully predicted the creation of various structures of colloidal particles, including clusters of Janus particles[32], polymerization of colloids[38], and the crystallization of multipatch colloidal particles[39]. Kern and Frenkel[40] introduced a simple model to describe the self-assembly of amphibolic particles with two-particle interactions. $\mathbf{r}_{ij}$ denotes a unit vector connecting the centers of particles $i$ and $j$, $r_{ij}$ is the corresponding distance, and $\mathbf{n}_i$ and $\mathbf{n}_j$ are unit vectors encoding the directions of patchy spheres. The Kern–Frenkel potential was defined as

$$U_{ij}^{KF} = u(r_{ij})\Omega(\mathbf{r}_{ij}, \mathbf{n}_i, \mathbf{n}_j), \qquad (24)$$

where

$$u(r_{ij}) = \begin{cases} \infty, & r_{ij} \le \sigma \\ -\epsilon, & \sigma < r_{ij} < \sigma + \Delta \\ 0, & r_{ij} > \sigma + \Delta. \end{cases}$$

and

$$\Omega(\mathbf{r}_{ij}, \mathbf{n}_i, \mathbf{n}_j) = \begin{cases} 1 & \text{if} \begin{cases} \mathbf{r}_{ij} \cdot \mathbf{n}_i > \cos\theta & \text{and} \\ \mathbf{r}_{ij} \cdot \mathbf{n}_j > \cos\theta \end{cases} \\ 0, & \text{otherwise.} \end{cases}$$

The characteristic quantity, $\chi = \sin^2(\theta/2)$, is the particle coverage. In the theory of self-assembly, the cluster-size distribution is determined by the partial partition functions Eq. (19). Due to the enormous number of possible configurations, it is impossible to calculate $\mathcal{Z}_j$ analytically and simulation methods were introduced, including a grand-canonical Monte Carlo method and successive umbrella sampling; for a review, see Rovigatti et al.[41]. Instead of calculating the exact value of $\mathcal{Z}_j$, we use a stylized model based on Fantoni et al.[32]. There the partial partition function is parameterized as $\frac{\log \mathcal{Z}_j}{j\epsilon} = b\tanh(aj)$, where $b < 0$ and $a > 0$ are the model parameters. While for small cluster sizes, the free energy per particle decreases linearly with the size, for larger clusters, it saturates at $b$. To calculate the average cluster size, Eq. (16), one has to solve the equation for $\Lambda$, Eq. (15). In Fig. 2, we show the phase diagram of the patchy particles for $b = -3$ and $a = 25$ and $n = 100$. The average number of clusters, $M$, plays the role of the order parameter. In the phase diagram, one can clearly distinguish three phases. At high temperature, we observe the liquid phase, where most particles are not bound to others. At low temperatures, we have a condensed phase with macroscopic clusters. The two phases are separated by a coexistence phase, where both large clusters and unbounded particles are present.

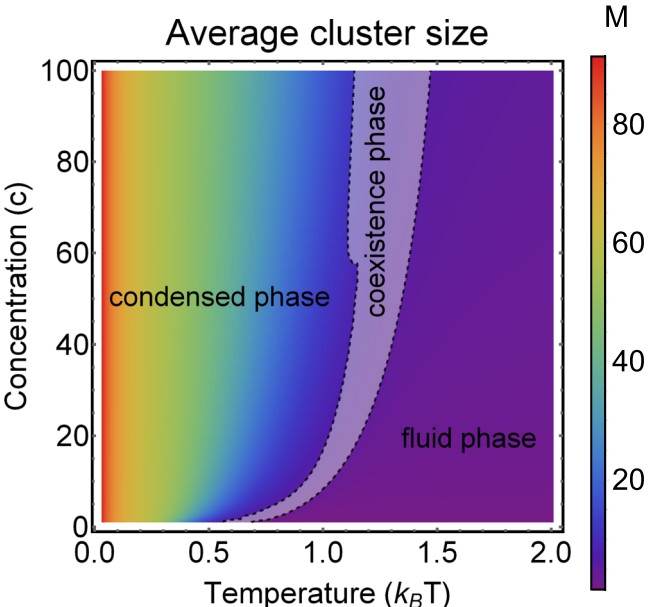

**Fig. 2 Phase diagram for the self-assembly of patchy particles for $n = 100$ particles.** The average cluster size ($M$) as a function of temperature ($T$) and concentration ($c$) is seen. The cluster size is given by the color and ranges from $M = 0$ (purple) to $M = 100$ (red). We observe three phases: the liquid and condensed phase are divided by a coexistence phase (gray area). Coexistence is characterized by a bimodal distribution that can be detected with a shift in the bimodality coefficient.

The coexistence phase (gray region) is characterized by a bimodal distribution that can be recognized by calculating the bimodality coefficient[42]. Results presented in Fig. 2 qualitatively correspond to results obatined in Fantoni et al.[32] for the case of Janus particles with $\chi = 0.5$.

*Curie–Weiss model with molecule formation.* To discuss an example of a spin system with molecule states, consider the fully connected Ising model[43–46] with a Hamiltonian that allows for possible molecule states

$$H(\sigma_i) = -\frac{J}{n-1}\sum_{i\ne j, \text{ free}} \sigma_i\sigma_j - h\sum_{j, \text{ free}} \sigma_j. \qquad (25)$$

Molecule states neither feel the spin–spin interaction nor the external magnetic field, $h$. Therefore, the sum only extends over free particles. In a mean-field approximation, we use the magnetization, $m = \frac{1}{n-1}\sum_{i\ne j}\sigma_i$, and express the Hamiltonian as $H^{MF}(\sigma_i) = -(Jm + h)\sum_{j,\text{free}}\sigma_j$. The self-consistency equation $m = -\frac{\partial F}{\partial h}\big|_{h=0}$ leads to an equation for $m$ that is calculated numerically (Supplementary Information) and that is shown in Fig. 3. Contrary to the mean-field approximation of the usual fully connected Ising model (without molecule states), the phase transition is no longer second-order but becomes first-order. There exists a bifurcation where solutions for $m = 0$ and $m > 0$ are stable. The second-order transition is recovered for small systems, $n \to 0$. The critical temperature is shifted toward zero for increasing $n$. We performed Monte Carlo simulations to check the result of the mean-field approximation; see Supplementary Information.

**Stochastic thermodynamics of structure-forming systems.** Consider an arbitrary nonequilibrium state given by $\wp_i^{(j)} \equiv \wp_i^{(j)}(t)$, and imagine that the evolution of the probability distribution is defined by a first-order Markovian linear master equation, as is usually

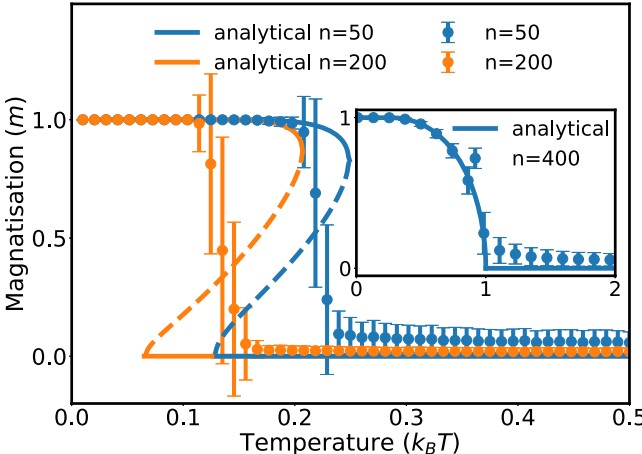

**Fig. 3 Magnetization of the fully connected Ising model with molecule states for $n = 50$ and $n = 200$ particles, for a spin–spin coupling constant, $J = 1$.** Results of the mean-field approximation (solid lines) are in good agreement with Monte Carlo simulations (symbols). Errorbars show the standard deviation of the average value obtained from 1000 independent runs of the simulations (see Supplementary Information for more details). The inset shows the well-known result for the fully connected Ising model without molecule states. Without molecule formation, we observe the usual second-order transition. With molecules, the critical temperature decreases with the number of particles and the phase transition becomes first-order.

assumed in stochastic thermodynamics[47,48]

$$\dot{\wp}_i^{(j)} = \sum_{kl} w_{ik}^{jl} \wp_k^{(l)} = \sum_{kl} \left( w_{ik}^{jl} \wp_k^{(l)} - w_{ki}^{lj} \wp_i^{(j)} \right). \tag{26}$$

$w_{ik}^{jl}$ are the transition rates. Note that probability normalization leads to $\sum_{ij} j \dot{\wp}_i^{(j)} = 0$. Given that detailed balance holds, $w_{ik}^{jl} \hat{\wp}_k^{(l)} = w_{ki}^{lj} \hat{\wp}_i^{(j)}$, the underlying stationary distribution, obtained from $\dot{\wp}_i^{(j)} = 0$, coincides with the equilibrium distribution Eq. (14). From this we get

$$\frac{w_{ik}^{jl}}{w_{ki}^{lj}} = \frac{j!}{l!} c^{l-j} \exp\left[ \alpha(l-j) + \beta\left( \epsilon_k^{(l)} - \epsilon_i^{(j)} \right) \right]. \tag{27}$$

The time derivative of the entropy per particle is

$$\frac{\mathrm{d}\mathcal{S}}{\mathrm{d}t} = -\sum_{ij} \dot{\wp}_i^{(j)} \log \wp_i^{(j)} - \sum_{ij} \dot{\wp}_i^{(j)} \log \left( \frac{j!}{c^{j-1}} \right). \tag{28}$$

Using the master Eq. (26) and some straightforward calculations, we end up with the usual second law of thermodynamics

$$\frac{\mathrm{d}\mathcal{S}}{\mathrm{d}t} = \dot{\mathcal{S}}_i + \beta \dot{\mathcal{Q}}, \tag{29}$$

where $\dot{\mathcal{Q}}$ is the heat flow per particle and $\dot{\mathcal{S}}_i$ is the nonnegative entropy production per particle, see Supplementary Information.

Let us now consider a stochastic trajectory, $\mathbf{x}(\tau) = (i(\tau), j(\tau))$, denoting that at time $\tau$, the particle is in state $x_{i(\tau)}^{(j(\tau))}$. We introduce the time-dependent protocol, $l(\tau)$, that controls the energy spectrum of the system. The stochastic energy for trajectory $\mathbf{x}$ $(\tau)$ and protocol $l(\tau)$ can be expressed as $\epsilon(\tau) \equiv \epsilon_{i(\tau)}^{(j(\tau))}(l(\tau))$. We assume microreversibility from which follows that detailed balance is valid even when the energy spectrum is time-dependent (due to protocol $l(\tau)$). We define the stochastic

entropy as

$$s(\mathbf{x}(\tau)) = -\left( \log \wp_{i(\tau)}^{(j(\tau))}(\tau) - 1 \right) - \log \left( \frac{j(\tau)!}{c^{j(\tau)-1}} \right). \tag{30}$$

We show that $\dot{s}(\mathbf{x}(\tau)) = \dot{s}_i(\mathbf{x}(\tau)) + \dot{s}_e(\mathbf{x}(\tau))$, where $\dot{s}_i$ is the stochastic entropy production rate and $\dot{s}_e$ is the entropy flow equal to $\dot{q}/T$, where $\dot{q}$ is the heat flow in Supplementary Information.

The time-reversed trajectory is $\tilde{\mathbf{x}}(\tau) = (i(T - \tau), j(T - \tau))$, and the time-reversed protocol is $\tilde{l}(\tau) = l(T - \tau)$. The log-ratio of the probability, $\mathcal{P}$, of a forward trajectory and the probability, $\tilde{\mathcal{P}}$, of the time-reversed trajectory under the time-reversed protocol is equal to $\Delta\sigma = \Delta s_i + \log \frac{j_0}{\tilde{j}_0}$, where $j_0 = j(\tau = 0)$ and $\tilde{j}_0 = \tilde{j}(\tau = 0)$, see Supplementary Information. Hence, $\log \frac{\mathcal{P}(\mathbf{x}(\tau))}{\tilde{\mathcal{P}}(\tilde{\mathbf{x}}(\tau))} = \Delta\sigma$, which leads to the fluctuation theorem[49]

$$\log \frac{P(\Delta\sigma)}{\tilde{P}(-\Delta\sigma)} = \Delta\sigma. \tag{31}$$

Assuming that the initial state is an equilibrium state, introducing the stochastic free energy, $f(\tau) = \epsilon(\tau) - Ts(\tau)$, and combining the first and the second law of thermodynamics, we get $\Delta s_i = \beta(w - \Delta f)$. The stochastic free energy of an equilibrium state is $f(\hat{\wp}_i^{(j)}) = -j\frac{\alpha}{\beta} - \frac{1}{\beta}$, see Supplementary Information.

If we start in an equilibrium distribution with $j(\tau = 0) = j_0$ and the reverse experiment also starts in an equilibrium distribution with $\tilde{j}(\tau = 0) = \tilde{j}_0$, by plugging this into Eq. (31) and a simple manipulation, we have

$$\frac{\mathcal{P}(\mathbf{x}(\tau)|j_0)}{\tilde{\mathcal{P}}(\tilde{\mathbf{x}}(\tau)|\tilde{j}_0)} = \exp\left( \beta w - \beta \left[ \Phi_{\tilde{j}_0}(\tilde{l}(0)) - \Phi_{j_0}(l(0)) \right] \right), \tag{32}$$

where $\Phi_j$ is the partial free energy Eq. (21). Finally, by a straightforward calculation, we obtain Crooks' fluctuation theorem[49,50]

$$\frac{P(w|j_0)}{\tilde{P}(-w|\tilde{j}_0)} = \exp(\beta(w - \Delta\Phi_j)) \tag{33}$$

where $\Delta\Phi_j = \Phi_{\tilde{j}_0}(\tilde{l}(0)) - \Phi_{j_0}(l(0))$. For technical details, see Supplementary Information.

## Discussion

We presented a straightforward way to establish the thermodynamics of structure-forming systems (e.g., molecules made from atoms or clusters of colloidal particles) based on the canonical ensemble with a modified entropy that is obtained by the proper counting of the system's configurations. The approach is an alternative to the grand-canonical ensemble that yields identical results for large systems. However, there are significant deviations that might have important consequences for small systems, where the interaction range becomes comparable with system size. Note that our results are valid for large systems (in the thermodynamic limit) as well as small systems at nanoscales. We showed that fundamental relations such as the second law of thermodynamics and fluctuation theorems remain valid for structure-forming systems. In addition, we demonstrated that the choice of a proper entropic functional has profound physical consequences. It determines, for example, the order of phase transitions in spin models.

We mention that we follow a similar reasoning as has been used in the case of Shannon's entropy: originally, Shannon's entropy was derived by Gibbs in the thermodynamic limit using a frequentist approach to statistics (probability is given by a large number of repetitions). However, once the formula for entropy had been derived, its validity was extended beyond the

thermodynamic limit, which corresponds to the Bayesian approach. It has been shown, e.g., by methods of stochastic thermodynamics, that the formula for the Shannon's entropy and the laws of thermodynamics remain valid for systems of arbitrary size (with the exception of systems with quantum corrections) and arbitrarily far from equilibrium[47]. In this paper, we follow the same type of reasoning for the case of structure-forming systems.

Typical examples where our results apply are chemical reactions at small scales, the self-assembly of colloidal particles, active matter, and nanoparticles. The presented results might also be of direct use for chemical nanomotors[51] and nonequilibrium self-assembly[35]. A natural question is how the framework can be extended to the well-known statistical physics of chemical reactions[23–26] where systems are composed of more than one type of atom.

## Data availability

Source Data are provided with this paper. All relevant data are available at: https://github.com/complexity-science-hub/Thermodynamics-of-structure-forming-systems.

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

## Acknowledgements

The authors acknowledge support from the Austrian Science fund Projects I 3073 and P 29252 and the Austrian Research Promotion agency FFG under Project 857136. The authors would like to thank Tuan Pham for helpful discussions.

## Author contributions

J.K., R.H., and S.T. conceptualized the work, S.D.L. performed the computational work, and all authors contributed to analytic calculations and wrote the paper.

## Competing interests

The authors declare no competing interests.
