## [Peer Review File · Nature Communications]

Reviewer #1 (Remarks to the Author):

Korbel, Linder, Hanel and Thurner

Thermodynamics of small systems with emergent structures.

Referee report by Henrik Jeldtoft Jensen

GENERAL REMARKS

The title is of course very interesting and will attract interest. The paper is enthusiastic but at the moment the content does not amount to more than a somewhat causal exposition of some preliminary ideas. I mean to say this in a friendly constructive manner. The authors and I have known each other for a long time and my hope is to inspire them to go through this first version in a careful way to help us understand some of the open and very fundamental problems touched upon in their manuscript.

The title gives the impression that the paper is an attempt to relate to thermodynamics. The reader will then expect two important properties of the considered entropy to be satisfied, namely additivity and extensivity. That thermodynamics implies these two properties were most rigorously demonstrated to be a consequence of the 2nd law by Lieb and Yngvason 'The Mathematical Structure of the Second law of Thermodynamics' in *Contemporary Developments in Mathematics 2001*, pp. 89–129, International Press (2002). The expression introduced in Eq. (3) is neither extensive nor additive. So, an explanation of how this "entropy" can be used for thermodynamic analysis is needed. Moreover, the authors point out that the functional in Eq. (3) is not maximised on the uniform distribution. That an entropy is maximised on the uniform distribution is often referred to as the 2nd Shannon-Khinchin axion and is considered to be a fundamental requirement in information theoretic uses of entropies. I suppose one may translate this into a Bayesian principle: if one doesn't know anything, everything is equally likely. Given these considerations, one may wonder in what sense S in Eq. (3) has anything to do with what is usually referred to as an entropy.

In this context it becomes relevant to pay a little bit attention to the content of Ref. [14]. This paper, which introduced the "emergent structure" model, the canonical ensemble (i.e. Max Ent under the constraint of average energy) is discussed in terms of the well-defined and well-behaved Group Entropies. These entropies are extensive, as the authors of course know very well, and composable, though not additive. The lack of additivity on the full probability space makes their connection to thermodynamics intriguing, but at least the extensivity ensures that a canonical ensemble can be established and that the derivative of the entropy with respect to the energy is intensive, allowing at least in principle for the definition of a thermodynamic temperature in the ordinary sense.

The lack of extensivity of the expression on Eq. (3) leads to complications later in the paper, most clearly seen in the lack of extensivity of the expressions in Eq (10) and (14) of the Supplementary material. The lack of extensivity of these formulae is particular puzzling, since they are derived using, what the authors call, the reduced sample space, i.e. $W \rightarrow W/n!$

SPECIFIC COMMENTS

It would be reasonable to explain to the reader why the procedures for establishing the relevant entropy for a complex system described in Ref. 2-5 are irrelevant to the case considered in the present paper.

It is of equal interest to the reader to know why the axiomatic group entropies, which allow the establishment of an extensive entropy maximised on the equal probability distribution, are irrelevant for the discussion. The relevant version of the group entropies is determined from the functional form of $W(n)$ and was already published in Ref. [14] and the algorithmic procedure for the computation of the entropy for a given $W(N)$ is described in

Piergiulio Tempesta and Henrik J Jensen, Universality Classes and Information-Theoretic Measure of Complexity via Group Entropies. *Scientific Reports* 10, 5952 (2020). doi.org/10.1038/s41598-020-60188-y

and

H.J. Jensen and P. Tempesta, Group Entropies: From Phase Space Geometry to Entropy Functionals via Group Theory, *Entropy* 2018, 20, 804; doi:10.3390/e20100804.

It is unclear what is meant by “temperature” in the two Figures 1 in the paper and in the Suppl. Mat. Since we are dealing with thermodynamics, we will expect the inverse temperature to equal the partial derivative of the entropy with respect to the energy. But, as mentioned above, the non-extensivity of the “entropy” given in Eq. (3) makes this derivative non-intensive, which is not really possible for a temperature.

All these problems are related to the so-called Gibbs paradox, which the authors do mention, but in a too casual manner. The authors correctly point out that Swendsen in Ref. [24] argues that one

does need to divide by $n!$ irrespectively of whether particles are distinguishable or not and of course there is a large literature on the $n!$ factor. But if dividing by $n!$ is the cure why are the F in Eq. ((1) and (14) in the Suppl. Mat. not extensive?

This kind of off the cuff casual approach may also be behind the statement line one left column. 2: "...many configurations represent the same micro-state". It seems to make more sense if the sentence were to refer to "macro-states".

In the same vein, the remark in [25] "See Suppl Mat. at *** for finer technical details, which include Refs. [26-32]." In my version of Suppl. Mat. there are no *** to be found and even if there was, the reader can only speculate which "finer mathematical details" are being referred to. The expression isn't exactly scientific precision and it could take a very long time to figure out where in the references [26-32] they are to be found.

Reviewer: Henrik Jeldtoft Jensen

Reviewer #2 (Remarks to the Author)

Dear editor,

I have read with attention the manuscript entitled “Thermodynamics of small systems with emergent structures” by Korbel et al and submitted for publication in Nature Communications.

After a short summary of the paper’s content I shall give my comments on the paper.

The article discusses the thermodynamic features of a system comprising many particles which can combine to form “molecules” with various numbers of particles. While such analysis is usually performed in the equilibrium grand canonical ensemble, the present paper proposes an equilibrium canonical ensemble treatment followed by an extension to non-equilibrium situations. The paper relies on combinatorics arguments for one of their main findings at equilibrium and on stochastic thermodynamics for the non-equilibrium extension of those results.

The paper addresses a technical problem in statistical mechanics which is surely worth investigating: a genuine canonical ensemble treatment of mixtures whose components can undergo chemical reactions. This is becoming more and more relevant indeed as experimental techniques enable us to probe smaller and smaller system sizes under various controlled environments. However, I felt that the importance and novel character of the work was not “fleshed out” strongly enough in the introduction and nor was due mention made of works in the physical chemistry literature on the matter either.

The paper is well written and in clear English but I have multiple issues with the clarity of the objectives, derivations and approximations being made.

For these two reasons I recommend rejection of the paper unless a very strong case be made by the authors. Please see below detailed comments about some aspects I found problematic with the paper.

Most of my comments will focus on the first, equilibrium part, which, if unclear or incorrect, jeopardises the validity of the paper’s findings in and out of equilibrium.

- 1- I will first mention that it was not clear at all to me what the authors called a “state” before Eq. (1) and the explanations in the following paragraph on page 2 did not help much. I believe that it is possible to clearly state, and formally express, what will constitute a “macrostate” for which we are going to derive the corresponding entropy.

- 2- Continuing from point 1, I felt that the part on combinatorics illustrated with some examples was confusing. First, I am actually not convinced by the general combinatorics arguments being provided. Second, there is missing information to make sense of the provided examples. If we have a system of four particles and each particle can be in two states, we still need to know what is the (macro)“state” of the system to understand the given example. In the first example I can only infer that it was $(n_1^{(1)} = 2, n_2^{(1)} = 2)$. As for the second example where the four particles can form molecules I am not sure what was the state here. In principle we would need to know how many states are available to a molecule and how many molecules are formed.
- 3- Following on from point 2, I do not understand why the degeneracy $W(\{n_i^{(j)}\})$ is not “just” a multinomial factor. Once a particle belongs to a j -molecule, the said molecule can be in one of various m_j states. Therefore I have difficulty appreciating the meaning of considering permutations of the j particles making up the molecule and giving rise to the factor $(j!)^{n_i^{(j)}}$ in Eq. (2). To close on this comment, I would have naïvely imagined that to form $n_i^{(j)}$ j -molecules in molecular state i , the combinatorial factor emerges from having to choose $jn_i^{(j)}$ particles among n from which to form them. Doing this for all possible molecules in the system would give the multinomial factor

$$W_{\text{alternative}}(\{n_i^{(j)}\}) = \frac{n!}{\prod_{j=1}^m \prod_{i=1}^{m_j} (jn_i^{(j)})!}$$

- 4- In spite of my previous comment, I am more than happy to consider that I am missing something and that Eq. (2) is actually valid. The problem is that, as far as my understanding is concerned, Eq. (3) does not follow from Eq. (2). The expression of Eq. (3) appears indeed quite natural from textbook statistical mechanics but it does not take into account the fact that $\sum_{ij} n_i^{(j)} \neq n$ as the authors have reminded the reader multiple times. So, taking the log of Eq.(2) should give:

$$S(\{n_i^{(j)}\}) = n \ln n - n - \sum_{ij} n_i^{(j)} \ln n_i^{(j)} + \sum_{ij} n_i^{(j)} - \sum_{ij} n_i^{(j)} \ln j!.$$

If my remark is correct then this poses some problems for the rest of the article’s findings.

- 5- Again, admitting that Eq. (3) holds as well, I actually take issue with the association of $p_i^{(j)}$ with some ‘probabilities’. Granted, the authors put quotation marks but then the term $-\sum_{ij} p_i^{(j)} \ln p_i^{(j)}$ cannot be interpreted as a form of entropy. One of the issues it may lead to is that since $p_i^{(j)}$ is not a probability (in the sense that it is not normalised when summing over i and j), then this ‘entropy’ looking term is likely to not satisfy Jensen’s inequality which is a hallmark feature of entropy-like quantities.

- 6- A further issue with regards to Eq. (3) is that there is no discussion on the validity of using Stirling's approximation for all $n_i^{(j)}$ while the paper claims to deal with small system sizes. Even if n is large there is no guarantee that $n_i^{(j)}$ is large for all i and j .
- 7- Moving on to the transition from Eq. (9) to Eq. (10), I have no technical objection to make but this still confuses me about the assumptions being used to ground the results of the paper. If Gibbs' statistical mechanics is to be used, then given the Hamiltonian in Eq. (9) we can write the partition function of the system as

$$Z(\beta, n) = \sum_{\{n_i^{(j)}\}} W(\{n_i^{(j)}\}) e^{-\beta H(\{n_i^{(j)}\})} = \sum_{\{n_i^{(j)}\}} e^{\ln W - \beta H(\{n_i^{(j)}\})},$$

and the free energy of the system is $F(\beta, n) = -T \ln Z(\beta, n)$. The summation above is over all possible configurations or (macro)states (e.g. one with no molecule at all and all particles in their ground state or one with n/m molecules in their ground state). If $\ln W$ is extensive and n is very large, one may want to approximate the sum above with the largest term in the sum. This leads to finding the configuration $\{\hat{n}_i^{(j)}\}$ which maximises the argument of the exponential (basically what the authors do in Eq.(10) and (11)). But in this case this is an *approximation* and not a fundamental principle. In particular, it is not clear that such an approximation holds for small system sizes.

- 8- Another problem that should be discussed is the meaning of $\hat{p}_i^{(j)}$ obtained from Eq. (12) if its value is not a rational. Indeed $p_i^{(j)}$ is defined as being $n_i^{(j)}/n$ where both numbers are assumed integers. How should $\hat{p}_i^{(j)}$ be interpreted warrants then an explanation.
- 9- To finish, I wish to discuss the $n \ln n$ term subtracted to the entropy to make it extensive. Dividing W by $n!$ actually gives an expression which takes values necessarily less than or equal to 1. If anything, this looks like undercounting. It is clear that Eq. (3) is super extensive in general but, given some of the comments above, it could be a signature of either some misunderstanding or some invalid derivations. For example we note that a configuration such that $n_i^{(j)} = 0$ for all $j > 1$ and with $m_1 = 2$ is equivalent to a system of n paramagnetic spins in absence of magnetic field. Eq. (3) happens to be extensive in that case but becomes super-extensive if $n \ln n$ is subtracted. The reason why Eq. (3) works in that case, is because the $p_i^{(j)}$ become then actual probabilities because values of j larger than one do not count (this can also be seen from Eq. (4) by setting $j=1$ in the n^{j-1} term). Thus, it does not appear sensible to subtract $n \ln n$ to the entropy in general for such systems.

To summarise, I believe that the assumptions and approximations made by the authors need to be formulated more clearly so that the soundness and validity of their findings can be unambiguously evaluated; which is not possible at the moment.

Reviewer #3 (Remarks to the Author):

Report on *Thermodynamics of small systems with emergent structures* by J. Korbel, S. D. Lindner, R. Hanel, and S. Thurner

The Authors present a new kind of statistics for systems at fixed temperature and total number of particles that can assemble into structures (molecules). In such a canonical-ensemble framework, the number of molecules comes out to play the role of a state variable, determined by the number of particles and by a Lagrange multiplier in a maximum entropy scheme with a general Hamiltonian form (Eq. 15). In particular, the number of molecules enters the Helmholtz free energy as shown in Eq. 16. It is shown that such statistics extends the grand-canonical formalism for relatively small number of particles, while the two approaches do provide coincident free energy per particle (at any temperature) as the number of particles increases. The new statistics leads to an interesting form for the system's entropy per particle (Eq. 8), which is then elaborated/specified to characterize some prototype cases. The emergence of a second-order transition in an instance of Ising model is intriguing.

In my opinion, the work is interesting and well written. The underlying idea is simple and natural, but to the best of my knowledge it was not developed so far. The outcomes are worth to be published, although there are some critical points to be better commented and justified. Specifically, the connection with stochastic thermodynamics (final part of the manuscript) should be better discussed at the formal level. Moreover, there are some issues in the Supplementary Material (SM) that have to be fixed. Please see my remarks below.

1) The part concerning the extension of Jarzynski's equality (JE), Eq. 23, and Crooks fluctuation theorem (CFT), Eq. 22, needs to be formalized. While in the rest of the paper the Authors follow a rigorous approach, it seems to me that here there are some important missing steps. Please consider my remarks below, strictly interrelated one each other:

a) When dealing with the JE and CFT, there is always a *driven* transformation performed on some controlled parameter(s) the system. What is/are the parameters that are changed in a controlled way in this kind of transformation (what the Authors call "process of getting from initial state A to the final state B")? This should be clearly stated. By looking at the derivation of Eq. 21 given in the SM, it appears that the energies of the molecule states are deterministically changed. So I suppose that this is the peculiar kind of external drive that the Authors have in mind. Is it correct? On the other hand, in the master equation Eq. 18, and also in its elaboration Eq. 17 of the SM, it seems that the transition rates are constant. The time-dependence of the energies of the molecule states and the constancy of the transition rates can be compatible one with the other only in special cases. This is a point that has to be properly commented.

b) The derivation is clear up to Eq. 21 for the entropy rate [apart from a technical point, see my remark 2)]. There is then a big gap between Eq. 21 (and Eq. 22 in the SM in which the work W does enter) and the extended forms of CFT and JE given in Eqs. 22 and 23. By adopting a master equation approach, all quantities are *average* (ensemble) quantities. In particular, the work W that the authors are considering is an *average* work. On the contrary, in the context of JE and CFT one deals with work as stochastic variable referred to the trajectory followed by the individual system in a specific transformation. In this sense, there is no evident way to pass from Eq. 21 to Eqs. 22 and 23. The only point where such a step is mentioned is the sentence after Eq. 22 in the SM: "We can apply this expression directly

to the Crooks fluctuation theorem and Jarzynski equality and obtain the formulas in the main text". This does not suffice. In the absence of a formal proof, or at least of some sound logical steps, the Authors should be cautious and explicitly present Eqs. 22 and 23 as conjectured expressions to be proved.

c) Still at the conjecture level, please consider the following viewpoint. The number M of molecules (structures) acts as a state variable at equilibrium in Eq. 16 which gives the Helmholtz free energy in terms of M . If I am correct, M behaves exactly as a quantity changeable and fixed through an external control on some other system's parameters (to be specified). Thus, I would expect that the CFT and JE hold in their standard forms (not Eqs. 22 and 23), where 'state A' corresponds to a number M_A and 'state B' to a number M_B , and where the switch on M is externally regulated in some way (to be figured out). This is an alternative viewpoint which would lead to very different results. What is wrong in my argument?

2) Maybe I have missed something, but the summation $\sum_{i,j} p_i$ should be equal to the number or molecules per particle, M/n ; hence $\sum_{i,j} \dot{p}_i = \dot{M}/n$. Why n does not appear in equations from Eq. 21 on?

3) The word "small", talking about systems, can be misleading. The Authors should better specify from beginning which is the scale of applicability of their theory. For instance, in the chemical ambit with "small systems" one normally means single nanostructures (which is the typical ambit of stochastic thermodynamics), of systems with tens of molecules (which is the ambit of stochastic kinetics). Here the Authors are more placed at the mesoscale of thousands of particles. A comment should also be made on the indicative threshold on the number of particles above which the approximations made are likely accurate (Stirling's formula and the approximation from Eq. 6 to Eq. 7): if one really goes down to "small systems" as intended in (bio)chemical ambits, I guess that these approximations degrade. In their conclusions, the Authors mention applicability 'chemical reactions at the small scale' and 'chemical nano-motors'. I would be cautious, please think carefully to the scale of applicability.

4) There are some technical issues to be checked/corrected in the SM. Please refer to the following piece of text that I have copied-pasted from the SM:

By plugging in the master equation we can further obtain that

$$\begin{aligned}
\dot{S} &= -\dot{M} - \sum_{ijkl} w_{ik}^{jl} p_k^{(l)} \log p_i^{(j)} - \sum_{ijkl} w_{ik}^{jl} p_k^{(l)} \left(\frac{j!}{c^{j-1}} \right) \\
&= -\dot{M} + \frac{1}{2} \sum_{ijkl} (w_{ki}^{lj} p_i^{(j)} - w_{ik}^{jl} p_k^{(l)}) \log \frac{p_i^{(j)}}{p_k^{(l)}} + \frac{1}{2} \sum_{ijkl} (w_{ki}^{lj} p_i^{(j)} - w_{ik}^{jl} p_k^{(l)}) \log \left(\frac{k!}{j!} c^{j-k} \right) \\
&= -\dot{M} + \frac{1}{2} \sum_{ijkl} \underbrace{(w_{ki}^{lj} p_i^{(j)} - w_{ik}^{jl} p_k^{(l)}) \log \frac{w_{ki}^{lj} p_i^{(j)}}{w_{ik}^{jl} p_k^{(l)}}}_{\dot{S}_i \geq 0} + \frac{1}{2} \sum_{ijkl} (w_{ki}^{lj} p_i^{(j)} - w_{ik}^{jl} p_k^{(l)}) \log \left(\frac{k!}{j!} c^{j-k} \frac{w_{ik}^{jl}}{w_{ki}^{lj}} \right) \\
&= -\dot{M} + \dot{S}_i + \underbrace{\frac{\beta}{2} \sum_{ijkl} (w_{ki}^{lj} p_i^{(j)} - w_{ik}^{jl} p_k^{(l)}) (\epsilon_k^{(l)} - \epsilon_i^{(j)})}_{\dot{S}_e = -\beta \dot{Q}} + \frac{\alpha}{2} \sum_{ijkl} (w_{ki}^{lj} p_i^{(j)} - w_{ik}^{jl} p_k^{(l)}) (l-j). \tag{17}
\end{aligned}$$

Let us note that from the first law of thermodynamics,

$$\dot{U} = \sum_{ij} \dot{p}_i^{(j)} \epsilon_i^{(j)} + \sum_{ij} p_i^{(j)} \dot{\epsilon}_i^{(j)} = \dot{Q} + \dot{W}, \tag{18}$$

the entropy flow is equal to the heat flow over the temperature. Let us focus on last term, which can be expressed as

$$\frac{1}{2} \sum_{ijkl} (w_{ki}^{lj} p_i^{(j)} - w_{ik}^{jl} p_k^{(l)}) (l-j) = \frac{1}{2} \sum_{ijkl} w_{ki}^{lj} p_i^{(j)} l - \frac{1}{2} \sum_{ijkl} w_{ki}^{lj} p_i^{(j)} j - \frac{1}{2} \sum_{ijkl} w_{ik}^{jl} p_k^{(l)} l + \frac{1}{2} \sum_{ijkl} w_{ik}^{jl} p_k^{(l)} j = \sum_{ij} \dot{p}_i^{(j)} j. \tag{19}$$

- Perhaps a \log is missing in the first line of Eq. 17.
- In the second line of Eq. 17, it seems that k should be replaced by l , please check. Please check also the sign $+$. Should it be $-$? In that case, correct the sign also the subsequent lines.
- In the third line of Eq. 17, the curly bracket should comprise the whole summation.
- In the last line of Eq. 17, please check the sign $-$ below the curly bracket (to be consistent with Eq. 20).
- In Eq. 19, the highlighted terms do not cancel each other. It seems that the first term cancels with the third, and the second with the fourth when one renames the indices in the summations. In practice, the sum in Eq. 19 gives directly zero without need of further comments.
- Please check if \dot{M} should be instead \dot{M}/n [my remark 2)].

Minor points

5) In the SM, please comment the fact that Eq. 10 and Eq. 15 do coincide for $c = 2$ which corresponds to the fractional $b = \frac{1}{2}$. What is the physical meaning of such a fact?

6) In several parts of the paper, and even in the title, the Authors talk about “emergent structures”. This is quite evocative, but it should be better explained what the word “emergent” means for the Authors.

7) The theme of “emergent structures” in a broad sense have attracted attention in the last years. In particular, is there some connection between the Authors’ framework and other ambits like that of J. England [see, e.g., *Nat. Nanotechnol.* **10**, 923 (2015)] ? I must say that the title of the manuscript suggests to be in the latter ambit, if the word ‘emergent’ is intended as a dynamic response in out-of-equilibrium conditions; but this is not the case.

Response to Referee 1

Reviewer: Henrik Jeldtoft Jensen

Dear Henrik, thank you very much for your inspiring comments that helped us to significantly improve the manuscript.

GENERAL REMARKS

The title is of course very interesting and will attract interest. The paper is enthusiastic but at the moment the content does not amount to more than a somewhat causal exposition of some preliminary ideas. I mean to say this in a friendly constructive manner. The authors and I have known each other for a long time and my hope is to inspire them to go through this first version in a careful way to help us understand some of the open and very fundamental problems touched upon in their manuscript.

The title gives the impression that the paper is an attempt to relate to thermodynamics. The reader will then expect two important properties of the considered entropy to be satisfied, namely additivity and extensivity. That thermodynamics implies these two properties were most rigorously demonstrated to be a consequence of the 2nd law by Lieb and Yngvason 'The Mathematical Structure of the Second law of Thermodynamics' in Contemporary Developments in Mathematics 2001, pp. 89–129, International Press (2002). The expression introduced in Eq. (3) is neither extensive nor additive. So, an explanation of how this "entropy" can be used for thermodynamic analysis is needed.

Response: In the improved manuscript, we show that the entropy introduced in the paper is indeed **additive**, according to the axioms used by Lieb and Yngvason. Moreover, for a constant concentration, it is also **extensive**. We mention this fact on p. 3, and finer technical details are elaborated in the SM.

Moreover, the authors point out that the functional in Eq. (3) is not maximised on the uniform distribution. That an entropy is maximised on the uniform distribution is often referred to as the 2nd Shannon-Khinchin axiom and is considered to be a fundamental requirement in information theoretic uses of entropies. I suppose one may translate this into a Bayesian principle: if one doesn't know anything, everything is equally likely. Given these considerations, one may wonder in what sense S in Eq. (3) has anything to do with what is usually referred to as an entropy.

Response: the reviewer is correct that the entropy **does not fulfill the second SK axiom**. However, there is a **good reason** for that. In the discussed case, not all states are of the same kind, and 'relabeling' the states might change the entropy. For example, if we denote free particles as molecules and molecules as free particles, the systems state is different, and the entropy changes. The validity of 2nd SK axiom relates to the requirement that the entropy is a symmetric function of probabilities. However, by this requirement, one also adds some knowledge (or an assumption) on the system. For example, by using Shannon entropy, one implicitly assumes that in equilibrium, subsystems are independent. While this is undoubtedly true for a large

class of systems, it is not right, e.g., entangled quantum particles or high-energy proton-proton collisions in CERN (see, e.g., Ref. [13] in the main text.) By assuming symmetry of the entropy, one considers that **only one type of state exists. This is certainly not true for structure-forming systems.**

From a philosophical point of view, if the only information about a system was its average energy, all entropic functionals are likely to be used. In practice, however, one knows from which system the data are obtained. If this system consists of molecules, then the entropy presented in the paper is an appropriate measure of uncertainty. One can reformulate the 'Bayesian principle' to the slightly different from that in equilibrium distribution; each microstate has a priori the same probability. This principle can then be translated to mesostates by proper calculation of multiplicity. We tackle this issue now in the SM.

In this context it becomes relevant to pay a little bit attention to the content of Ref. [14]. This paper, which introduced the "emergent structure" model, the canonical ensemble (i.e. Max Ent under the constraint of average energy) is discussed in terms of the well-defined and well-behaved Group Entropies. These entropies are extensive, as the authors of course know very well, and composable, though not additive. The lack of additivity on the full probability space makes their connection to thermodynamics intriguing, but at least the extensivity ensures that a canonical ensemble can be established and that the derivative of the entropy with respect to the energy is intensive, allowing at least in principle for the definition of a thermodynamic temperature in the ordinary sense.

Response: We added a discussion about various axiomatic schemes to page 3 and a whole new section in the SM, discussing the connection to various axiomatic systems. Since the entropy is additive and extensive (of course only for a constant concentration), the **temperature is also well-defined**. Moreover, we show that entropy is **group-composable**, in exactly the sense of Ref. [14]. However, we note that it does not belong to the class of Z-entropies, as proposed by Tempesta because the entropy does not fulfill the 2nd SK axiom. Moreover, the physical interpretation is explicitly demonstrated by the **validity of the 2nd law of thermodynamics**. Therefore, one can conclude that the Lagrange multiplier β **does indeed correspond to physical inverse temperature**.

The lack of extensivity of the expression on Eq. (3) leads to complications later in the paper, most clearly seen in the lack of extensivity of the expressions in Eq (10) and (14) of the Supplementary material. The lack of extensivity of these formulae is particularly puzzling, since they are derived using, what the authors call, the reduced sample space, i.e. $W \rightarrow W/n!$

Response: we agree that the connection to the GC ensemble was puzzling. Therefore, we omitted the discussion about the Gibbs paradox and the (in)distinguishability of particles and demonstrate the convergence to the GC ensemble by plotting specific heat, which is a physical quantity, independent from choosing W or $W/n!$. We leave a thorough discussion about the Gibbs paradox for future work.

SPECIFIC COMMENTS

It would be reasonable to explain to the reader why the procedures for establishing the relevant entropy for a complex system described in Ref. 2-5 are irrelevant to the case considered in the present paper.

Response: procedures established in Refs. [2-5] are **relevant** for our case, which we demonstrate by the calculation of the (c,d)-exponents for the case of entropy (3). However, the class of entropies established in Refs. [2-5] assume the 2nd SK axiom to hold, and therefore the entropy does **not** fall into the class of entropic functionals derived there.

It is of equal interest to the reader to know why the axiomatic group entropies, which allow the establishment of an extensive entropy maximised on the equal probability distribution, are irrelevant for the discussion. The relevant version of the group entropies is determined from the functional form of $W(n)$ and was already published in Ref. [14] and the algorithmic procedure for the computation of the entropy for a given $W(N)$ is described in

Piergiulio Tempesta and Henrik J Jensen, Universality Classes and Information-Theoretic Measure of Complexity via Group Entropies. Scientific Reports 10, 5952 (2020). doi.org/10.1038/s41598-020-60188-y

and

H.J. Jensen and P. Tempesta, Group Entropies: From Phase Space Geometry to Entropy Functionals via Group Theory, Entropy 2018, 20, 804; doi:10.3390/e20100804.

Response: as mentioned above, we show that the entropy is **group-composable** but does not belong to the class of entropies derived in the aforementioned references because they assume the 2nd SK axiom to hold.

It is unclear what is meant by “temperature” in the two Figures 1 in the paper and in the Suppl. Mat. Since we are dealing with thermodynamics, we will expect the inverse temperature to equal the partial derivative of the entropy with respect to the energy. But, as mentioned above, the non-extensivity of the “entropy” given in Eq. (3) makes this derivative non-intensive, which is not really possible for a temperature.

Response: the temperature is equal to the partial derivative of entropy w.r.t. energy. We have shown that for a constant concentration, entropy is **extensive**, and therefore, the temperature is **intensive**. However, for the case where the concentration is **not** constant, the temperature is not intensive because the rescaling of a system changes the microscopic structure of the system. That’s why we now call the system ‘structure-forming’.

All these problems are related to the so-called Gibbs paradox, which the authors do mention, but in a too casual manner. The authors correctly point out that Swendsen in Ref. [24] argues that one does need to divide by $n!$ irrespectively of whether particles are distinguishable or not and of course there is a large literature on the $n!$

factor. But if dividing by $n!$ is the cure why are the F in Eq. ((1) and (14) in the Suppl. Mat. not extensive?

Response: we agree that the discussion about the so-called Gibbs paradox was not very illustrative. In the current version, we omit the discussion and demonstrate the convergence to the GC ensemble by plotting the specific heat, which is independent of the presence of the $n!$ factor.

This kind of off the cuff casual approach may also be behind the statement line one left column. 2: "...many configurations represent the same micro-state". It seems to make more sense if the sentence were to refer to "macro-states".

Response: thank you, we have changed this part and use the term "mesostate" to avoid potential confusion with the thermodynamic macrostate described by thermodynamic quantities such as temperature, volume, internal energy, etc. – see also our response to Referee 2.

In the same vein, the remark in [25] "See Suppl Mat. at *** for finer technical details, which include Refs. [26-32]." In my version of Suppl. Mat. there are no *** to be found and even if there was, the reader can only speculate which "finer mathematical details" are being referred to. The expression isn't exactly scientifically precise and it could take a very long time to figure out where in the references [26-32] they are to be found.

Response: sorry for this misunderstanding. The reference to the supplementary material was a left-over from a previous version, and it should point to the supplementary material provided to this paper.

Thank you again for your great comments that helped us improve the manuscript, such that much confusion will be avoided in the present form.

List of changes that emerged from comments of Referee 1

- **We added a paragraph discussing connection of the structure-forming entropy to several axiomatic schemes (page 3 and SM).**
- **In comparison with the grand canonical ensemble, we omitted the discussion about Gibbs paradox altogether and used the specific heat for a comparison.**

Response to referee 2

First of all, we are sorry that the original report is now missing some of the formulas that we could not import to this document. For the sake of readability for the referee, the original PDF of the report is attached to the resubmission.

The article discusses the thermodynamic features of a system comprising many particles which can combine to form “molecules” with various numbers of particles. While such analysis is usually performed in the equilibrium grand canonical ensemble, the present paper proposes an equilibrium canonical ensemble treatment followed by an extension to nonequilibrium situations. The paper relies on combinatorics arguments for one of their main findings at equilibrium and on stochastic thermodynamics for the non-equilibrium extension of those results.

The paper addresses a technical problem in statistical mechanics which is surely worth investigating: a genuine canonical ensemble treatment of mixtures whose components can undergo chemical reactions. This is becoming more and more relevant indeed as experimental techniques enable us to probe smaller and smaller system sizes under various controlled environments. However, I felt that the importance and novel character of the work was not “fleshed out” strongly enough in the introduction and nor was due mention made of works in the physical chemistry literature on the matter either.

Response: we added a sentence to the introduction where we stress that the canonical approach to chemical reaction systems has not yet been considered to our knowledge.

The paper is well written and in clear English but I have multiple issues with the clarity of the objectives, derivations and approximations being made. For these two reasons I recommend rejection of the paper unless a very strong case be made by the authors. Please see below detailed comments about some aspects I found problematic with the paper.

Most of my comments will focus on the first, equilibrium part, which, if unclear or incorrect, jeopardises the validity of the paper’s findings in and out of equilibrium.

1- I will first mention that it was not clear at all to me what the authors called a “state” before Eq. (1) and the explanations in the following paragraph on page 2 did not help much. I believe that it is possible to clearly state, and formally express, what will constitute a “macrostate” for which we are going to derive the corresponding

entropy.

Response: thank you. We now added a formal definition of a **microstate** of a particle in a system of n particles. We have also added a formal definition of a **mesostate**, from which we later calculate the entropy. We hope that in the current version of the manuscript it is much clearer how we define states. We used the term “mesostate” to avoid potential confusion with the thermodynamic macrostate, described by thermodynamic variables (e.g. temperature, volume, internal energy, etc.).

2- Continuing from point 1, I felt that the part on combinatorics illustrated with some examples was confusing. First, I am actually not convinced by the general combinatorics arguments being provided. Second, there is missing information to make sense of the provided examples. If we have a system of four particles and each particle can be in two states, we still need to know what is the (macro)“state” of the system to understand the given example. In the first example I can only infer that it was

Response: by adding a proper definition of a microstate and a mesostate and explicitly writing down examples of the microstates corresponding to a given mesostate we hope that the approach should be clearer.

As for the second example where the four particles can form molecules I am not sure what was the state here. In principle we would need to know how many states are available to a molecule and how many molecules are formed.

Response: with the now precise definitions of a microstate and mesostate, the presented examples should now be clear and apparent.

3- Following on from point 2, I do not understand why the degeneracy $W(\{n\% (\&)\})$ is not “just” a multinomial factor. Once a particle belongs to a j -molecule, the said molecule can be in one of various $m\&$ states. Therefore I have difficulty appreciating the meaning of considering permutations of the j particles making up the molecule and giving rise to the factor $(j!)!$ in Eq. (2). To close on this comment, I would have naïvely imagined that to form j -molecules in molecular state i , the combinatorial factor emerges from having to choose jn - ($\&$) particles among n from which to form them. Doing this for all possible molecules in the system would give the multinomial factor.

Response: we have improved the description in the main text. We hope that the formula for the multiplicity is now more comfortable to follow and that the presented examples illustrate it clearly.

4- In spite of my previous comment, I am more than happy to consider that I am missing something and that Eq. (2) is actually valid. The problem is that, as far as my understanding is concerned, Eq. (3) does not follow from Eq. (2). The expression of Eq. (3) appears indeed quite natural from textbook statistical mechanics but it does not take into account the fact that

as the authors have reminded the reader multiple times. So, taking the log of Eq.(2) should give:
If my remark is correct then this poses some problems for the rest of the article's findings.

Response: here we thank the referee a lot for pointing out that the approximation was not good enough. By considering the second term in the Stirling approximation, **we have solved this issue** now. While the form of entropic functional and the MaxEnt distribution have slightly changed, the main results, such as the relation for the **Helmholtz free energy**, the form of the **2nd law of thermodynamics**, the **fluctuation theorems** and the presence of the **first-order transition** in the fully connected Ising model, all **remain unchanged**.

5- Again, admitting that Eq. (3) holds as well, I actually take issue with the association of p_{ij} with some 'probabilities'. Granted, the authors put quotation marks but then the term

cannot be interpreted as a form of entropy. One of the issues it may lead to is that since p_{ij} is not a probability (in the sense that it is not normalised when summing over i and j), then this 'entropy' looking term is likely to not satisfy Jensen's inequality which is a hallmark feature of entropy-like quantities.

Response: we added a representation of entropy in terms of real $\tilde{p}_i = \sum_j p_{ij}$, which is formally more appropriate. The entropy can be therefore expressed in terms of an **ordinary probability distribution** that adds up to one. Moreover, we now show that the entropy is indeed a concave function of all its arguments, which implies the Jensen's inequality.

6- A further issue with regards to Eq. (3) is that there is no discussion on the validity of using Stirling's approximation for all n_{ij} while the paper claims to deal with small system sizes. Even if n is large there is no guarantee that n_{ij} is large for all i and j .

Response: in the derivation of the entropic functional, we follow the classic approach of Gibbs and consider that the system is in the thermodynamic limit, i.e., $n_i \rightarrow \infty$. Here, the validity of the Stirling approximation (in the form that the referee suggested) should be granted. Note that later in the text, the validity of the formula is extended to small systems and for far-from-equilibrium systems since the second law of thermodynamics holds for these systems as well. **We added a discussion on the validity of our approach to the final part of conclusions** (see also the comments of Reviewer 3).

7- Moving on to the transition from Eq. (9) to Eq. (10), I have no technical objection to make but this still confuses me about the assumptions being used to ground the results of the paper. If Gibbs' statistical mechanics is to be used, then given the Hamiltonian in Eq. (9) we can write the partition function of the system as

and the free energy of the system is $F(\beta, n) = -T \ln Z(\beta, n)$. The summation above

is over all possible configurations or (macro)states (e.g. one with no molecule at all and all particles in their ground state or one with n/m molecules in their ground state). If $\ln W$ is extensive and n is very large, one may want to approximate the sum above with the largest term in the sum. This leads to finding the configuration

which maximises the argument of the exponential (basically what the authors do in Eq.(10) and (11)). But in this case this is an *approximation* and not a fundamental principle. In particular, it is not clear that such an approximation holds for small system sizes.

Response: here we need to note that the notion of the partition function in the form used by the referee is strictly constrained to **Shannon entropy**. As demonstrated by Eq. (17), Helmholtz free energy is defined as, $F = -\alpha/\beta - M/\beta$. The Lagrange multiplier α is given by Eq. (17), and therefore, free energy cannot be expressed as $-T \text{Log } Z$. Note that, however, for molecules of the same order, it is possible to introduce a **partial partition function**, which is defined expectedly (see the definition between Eq. (13) and Eq. (14)).

8- Another problem that should be discussed is the meaning of p_{i^j} obtained from Eq.(12) if its value is not a rational. Indeed p_{i^j} is defined as being n_{i^j}/n where both numbers are assumed integers. How should p_{i^j} be interpreted warrants then an explanation.

Response: since the equilibrium distribution is calculated in the thermodynamic limit, where both, $n \rightarrow \infty$, and $n_{i^j} \rightarrow \infty$, the ratio n_{i^j}/n can be **any real number** (between 0 and $1/j$). This is analogous to the case of Shannon entropy.

9- To finish, I wish to discuss the $n \ln n$ term subtracted to the entropy to make it extensive. Dividing W by $n!$ actually gives an expression which takes values necessarily less than or equal to 1. If anything, this looks like undercounting. It is clear that Eq. (3) is super extensive in general but, given some of the comments above, it could be a signature of either some misunderstanding or some invalid derivations. For example we note that a configuration such that $n\%$ and with $m! = 2$ is equivalent to a system of n paramagnetic spins in absence of magnetic field. Eq. (3) happens to be extensive in that case but becomes superextensive if $n \ln n$ is subtracted. The reason why Eq. (3) works in that case, is because the $p\%$ (&) become then actual probabilities because values of j larger than one do not count (this can also be seen from Eq. (4) by setting $j=1$ in the $n\&0!$ term). Thus, it does not appear sensible to subtract $n \ln n$ to the entropy in general for such systems.

Response: The referee is correct that the current explanation was not very illustrative. The only reason for subtracting $n \log n$ was the comparison with the GC ensemble, where this subtraction is done due to the Gibbs paradox. We have **omitted** the part on the Gibbs paradox and omitted the introduction of the reduced sample space. Instead, we compared the **specific heat** obtained by both approaches, which is *independent* of the choice between original and reduced

sample space. Moreover, it is a **physical, directly measurable quantity**. We think that this led to a much clearer presentation.

To summarise, I believe that the assumptions and approximations made by the authors need to be formulated more clearly so that the soundness and validity of their findings can be unambiguously evaluated; which is not possible at the moment

Response: Thank you very much for your comments. They helped us to improve the manuscript.

List of changes emerged from comments by Referee 2:

- **We added a formal definition of a system's micro-state and meso-state (pages 1-2).**
- **We improved examples illustrating multiplicity of various meso-states (page 2).**
- **We used a precise version of Stirling approximation (page 2). This resulted in a change of the entropic functional (pages 2-3) and the maxent distribution (page 4).**
- **We added a representation of entropy in terms of ordinary probability distributions, summing up to one (page 3).**
- **We showed that the entropy is indeed a concave function and therefore fulfills Jensen's inequality (page 4).**
- **We defined the partial partition function that appears when solving the normalization condition for the probability distribution (page 4).**

Response to Referee 3

The Authors present a new kind of statistics for systems at fixed temperature and total number of particles that can assemble into structures (molecules). In such a canonical ensemble framework, the number of molecules comes out to play the role of a state variable, determined by the number of particles and by a Lagrange multiplier in a maximum entropy scheme with a general Hamiltonian form (Eq. 15). In particular, the number of molecules enters the Helmholtz free energy as shown in Eq. 16. It is shown that such statistics extends the grand-canonical formalism for relatively small number of particles, while the two approaches do provide coincident free energy per particle (at any temperature) as the number of particles increases. The new statistics leads to an interesting form for the system's entropy per particle (Eq. 8), which is then elaborated/specified to characterize some prototype cases. The emergence of a second-order transition in an instance of Ising model is intriguing.

In my opinion, the work is interesting and well written. The underlying idea is simple and natural, but to the best of my knowledge it was not developed so far. The outcomes are worth to be published, although there are some critical points to be better commented and justified. Specifically, the connection with stochastic thermodynamics (final part of the manuscript) should be better discussed at the formal level. Moreover, there are some issues in the Supplementary Material (SM) that have to be fixed. Please see my remarks below.

1) The part concerning the extension of Jarzynski's equality (JE), Eq. 23, and Crooks fluctuation theorem (CFT), Eq. 22, needs to be formalized. While in the rest of the paper the Authors follow a rigorous approach, it seems to me that here there are some important missing steps. Please consider my remarks below, strictly interrelated one each other:

Response: The referee is correct. The derivation missed some logical points and was too sloppy. In the revised version of the manuscript, we **added the main points of the derivation** to the main text, while **finer technical details** are discussed in the supplementary material.

- a) When dealing with the JE and CFT, there is always a *driven* transformation performed on some controlled parameter(s) the system. What is/are the parameters that are changed in a controlled way in this kind of transformation (what the Authors call "process of getting from initial state A to the final state B")? This should be clearly stated. By looking at the derivation of Eq. 21 given in the SM, it appears that the energies of the molecule states are deterministically changed. So I suppose that this is the peculiar kind of external drive that the Authors have in mind. Is it correct? On the other hand, in the master equation Eq. 18, and also in its elaboration Eq. 17 of the SM, it seems that the transition rates are constant. The time-dependence of the energies of the molecule states and the constancy of the transition rates can be compatible one with the other only in special cases. This is a point that has to be properly commented.

Response: As the referee correctly mentioned, the control of the system is given by the **external driving** of the energy spectrum, now denoted as a protocol $I(\tau)$. In the SM, explicit labeling of the time-dependence of the energy spectrum is omitted in order to avoid too complicated a notation.

- b) The derivation is clear up to Eq. 21 for the entropy rate [apart from a technical point, see my remark 2)]. There is then a big gap between Eq. 21 (and Eq. 22 in the SM in which the work W does enter) and the extended forms of CFT and JE given in Eqs. 22 and 23. By adopting a master equation approach, all quantities are *average* (ensemble) quantities. In particular, the work W that the authors are considering is an *average* work. On the contrary, in the context of JE and CFT one deals with work as stochastic variable referred to the trajectory followed by the individual system in a specific transformation. In this sense, there is no evident way to pass from Eq. 21 to Eqs. 22 and 23. The only point where such a step is mentioned is the sentence after Eq. 22 in the SM: "We can apply this expression directly to the Crooks fluctuation theorem and Jarzynski equality and obtain the formulas in the maintext". This does not suffice. In the absence of a formal proof, or at least of some sound logical steps, the Authors should be cautious and explicitly present Eqs. 22 and 23 as conjectured expressions to be proved.

Response: Again, the referee is correct; the notation was not accurate. In the current version of the main text, we introduce stochastic (or trajectory) versions of thermodynamic quantities. We explicitly mention that the work appearing in the fluctuation theorems is a stochastic quantity.

- c) Still at the conjecture level, please consider the following viewpoint. The number M of molecules (structures) acts as a state variable at equilibrium in Eq. 16 which gives the Helmholtz free energy in terms of M . If I am correct, M behaves exactly as a quantity changeable and fixed through an external control on some other system's parameters (to be specified). Thus, I would expect that the CFT and JE hold in their standard forms (not Eqs. 22 and 23), where 'state A' corresponds to a number M_A and 'state B' to a number M_B , and where the switch on M is externally regulated in some way (to be figured out). This is an alternative viewpoint which would lead to very different results. What is wrong in my argument?

Response: this is indeed an interesting idea. Suppose one considers a protocol that changes the number of molecules, which can be, in practice done by presence of a particle reservoir. In that case, one can define the protocol in terms of a controlled chemical potential. The total work is the sum of the mechanical work and chemical work, and one recovers the results obtained by the grand-canonical ensemble. (where M can be understood as proportional to chemical work). We now briefly comment on this approach after the derivation of fluctuation theorems.

2) Maybe I have missed something, but the summation $\sum_{i,j} p_i$ should be equal to the number of molecules per particle, M/n ; hence $\sum_{i,j} p_i = M/n$. Why n does not appear in equations from Eq. 21 on?

Response: The notation was not very clear, we agree. In the new version of the manuscript, we use a **calligraphic** font for quantities per particle and a **normal** font for total quantities.

3) The word “small”, talking about systems, can be misleading. The Authors should better specify from beginning which is the scale of applicability of their theory. For instance, in the chemical ambit with “small systems” one normally means single nanostructures (which is the typical ambit of stochastic thermodynamics), of systems with tens of molecules (which is the ambit of stochastic kinetics). Here the Authors are more placed at the mesoscale of thousands of particles. A comment should also be made on the indicative threshold on the number of particles above which the approximations made are likely accurate (Stirling’s formula and the approximation from Eq. 6 to Eq. 7): if one really goes down to “small systems” as intended in (bio)chemical ambits, I guess that these approximations degrade. In their conclusions, the Authors mention applicability ‘chemical reactions at the small scale’ and ‘chemical nano-motors’. I would be cautious, please think carefully to the scale of applicability.

Response: We added a **discussion to the final part of conclusions** on the validity of the results concerning the system’s scale. The main result is just a repetition of results obtained for the case of Shannon (or Boltzmann-Gibbs) entropy, just for a different class of systems. Originally, Shannon entropy was of course derived by Gibbs for the case of equilibrium systems in the thermodynamic limit. However, it turned out that the validity of the entropic formula is broader. It has been shown, e.g., by the methods of stochastic thermodynamics, that the entropy is valid for small systems arbitrarily far from equilibrium. We used the similar reasoning for case of structure-forming systems. Similarly, in the current version of the manuscript, we have used the same version of Stirling’s approximation that is typically used for Shannon entropy. It is typically assumed that the formula works quite well for $n \approx 10^1$. Note that thanks to Referee 2, we now use a more accurate formula for Stirling’s approximation.

4) There are some technical issues to be checked/corrected in the SM. Please refer to the following piece of text that I have copied-pasted from the SM:

- Perhaps a *log* is missing in the first line of Eq. 17.
- In the second line of Eq. 17, it seems that k should be replaced by l , please check. Please check also the sign +. Should it be – ? In that case, correct the sign also the subsequent lines.
- In the third line of Eq. 17, the curly bracket should comprise the whole summation.
- In the last line of Eq. 17, please check the sign – below the curly bracket (to be consistent with Eq. 20).
- In Eq. 19, the highlighted terms do not cancel each other. It seems that the first term cancels with the third, and the second with the fourth when one renames the indices in the summations. In practice, the sum in Eq. 19 gives directly zero without need of further comments.
- Please check if M' should be instead M'/n [my remark 2)].

Response: Thank you. We updated this part of the SM accordingly.

Minor points

5) In the SM, please comment the fact that Eq. 10 and Eq. 15 do coincide for $c = 2$ which corresponds to the fractional $b = 1/2$. What is the physical meaning of such a fact?

Response: There was a typo, $c=n/b$, so we have that $b(n)=n/2$. We have commented this situation in the end of the derivation.

6) In several parts of the paper, and even in the title, the Authors talk about “emergent structures”. This is quite evocative, but it should be better explained what the word “emergent” means for the Authors.

7) The theme of “emergent structures” in a broad sense have attracted attention in the last years. In particular, is there some connection between the Authors’ framework and other ambits like that of J. England [see, e.g., *Nat. Nanotechnol.* **10**, 923 (2015)] ? I must say that the title of the manuscript suggests to be in the latter ambit, if the word ‘emergent’ is intended as a dynamic response in out-of-equilibrium conditions; but this is not the case

Response: as the reviewer correctly pointed out, the term *emergent* has several different meanings in the current literature. To avoid confusion, e.g., with the aforementioned work, we replaced the word emergent by the word **structure-forming**.

Thank you again very much for your helpful comments.

List of changes emerged from comments by Referee 3:

- We added the main points of the derivation of the fluctuation theorem to the main text (page 5). Technical details of the derivation have been added to the supplementary material.
- We added a discussion on the validity of the presented approach, especially we discussed applicability to systems at nanoscales.
- We improved the derivation of the second law of thermodynamics in the supplementary material.
- We replaced the possibly misleading word “emergent” by word “structure-forming” – also in the title.

Reviewers' comments:

Reviewer #2 (Remarks to the Author):

I have read with attention the revised version of the manuscript formerly entitled "Thermodynamics of small systems with emergent structures" by Korbelt et al and submitted for publication in Nature Communications.

I am thankful to the authors for having followed some of my recommendations.

The paper is now very much clearer than in its previous form; in particular the definition of a microstate and a macrostate have been greatly improved and the combinatorial reasoning is consequently more robust.

Nevertheless, I still hold concerns with regards to certain points:

I think there a typo in the normalisation condition left column page 2 before Eq. (1).

Before the green text page 3, it is said that the results are valid for any $b(n)$. I am not sure what results are referred to there. One of the needed properties of entropy, extensivity, is shown to work only with $b \propto n$ in Eq. (6) of the SM. If $b(n)$ can be anything I am not certain that the concept of "entropy per particle" is well defined.

Bottom of page 4 it is stated that "we do not restrict ourselves to the case of thermodynamic limit". Here again, I am not sure what is referred to. The Eqs for $\hat{p}_i^{(j)}$ substituted into Eq.(19) do come from derivations relying on the thermodynamic limit. Likewise, Eq. (20) comes from differentiation of Eq. (9) which uses the thermodynamic limit.

In fact Eq. (9) has been revised in the new version but its consequences for Eq. (20) have not been carried out. So, unless I am missing something, some extra terms should appear in Eq. (20) and maybe elsewhere too.

I am a bit puzzled by the interpretation of Eq. (28) and the derivation leading to it. The thermodynamic identities being used as well as the reasoning appear to imply that f and m can be considered as independent state variables with respect to which one can express entropy variations. This leads me to interpret the authors' work as showing that the canonical ensemble for n particles forming molecules is equivalent to a sub-ensemble with an additional fixed parameter: the number of molecules per particle, in the thermodynamic limit. This new ensemble is then associated to a brand new thermodynamic potential being $\beta F-M$ which explicitly appears in the exponent in Eq.(28). Whether I am correct or not, I felt that the manuscript was not clear on this matter.

The examples provided Fig1 and Fig2 seem very interesting but they appear totally overlooked and are only mentioned in passing. In that respect the presentation of the work and its impact seems fairly unbalanced.

Following the previous note, although interesting, I have the regret to say that the undertaken work and reporting of said work do not appear to me impactful enough to warrant publication in Nature Communication.

From the concerns listed above, I cannot recommend publication in Nature Communication and would suggest a more specialised journal.

Report on ***Thermodynamics of small structure-forming systems*** by J. Korbelt, S. D. Lindner, R. Hanel, and S. Thurner

The Authors has made a big effort to improve their manuscript according to the criticisms. I think that the exemplifications in the introduction, and the sections added in the Supplementary Material (SM), strengthen the argumentation a lot. Certainly the subject is very technical and faces the statistics/thermodynamics of structure-forming systems from different angles (equilibrium properties and out-of-equilibrium stochastic realizations). The same reader might hardly appreciate the whole work; conversely, specialists might have viewpoints against some of the contents, but undoubtedly this work makes thinking and could originate a lively debate.

Concerning my remarks, the Authors have commented and responded to all issues with appropriate modification/extension of the text. Many thanks for this. However, there are still a few points that have to be checked and possibly corrected prior publication. The main points are 1) and 2) below; the others are minor points. Once the Authors will have considered these issues, I can recommend the publication of this article in *Nature Communications*.

1) In the out-of-equilibrium part, I think that more precise statements are due in some points.

- When they introduce the master equation Eq. (18), I presume that the Authors are already thinking in all generality to a driven out-of-equilibrium system (i.e., such master equation is not restricted to a simple free relaxation with constant rates w_{ik}^d). If this is the case, the sentence “Assuming detailed balance such that (...)” should be rephrased by referring to the “underlying distribution” that would be attained by free relaxation at stopped protocol. Something like:

Assuming detailed balance even under out-of-equilibrium, that is [equation], requires that the underlying stationary distribution coincides with the distribution obtained from the maximum entropy principle. From this (...)

Do I have interpreted correctly your reasoning?

In this way, Eq. (18) could describe either a free relaxation or the evolution with driving in which the Markov process is non-stationary and the transition rates are time-dependent.

- Before Eq. (22) the Authors write ” Let us still consider that the detailed balance is fulfilled even for the time-dependent energies”. It seems more appropriate to say “Let us assume that (...)”, as above. This is indeed a widely accepted *assumption* in stochastic thermodynamics, more than a physical fact certainly true. Indeed, the validity of this assumption could be even more questionable here, where the transitions occur between very different physical states (the particles states) and not between structurally vicinal microstates as it occurs in the typical applications where the system is a manipulated single molecule... Moreover, I think it would more appropriate to refer to ‘microreversibility’, rather than to ‘detailed balance’ (of course

the two concepts are interrelated). This would be more adherent to the derivation of the fluctuation theorem given in the SM.

2) In the derivation of the fluctuation theorem given in the SM, there are some issues that have to be considered.

- Before introducing the Eq. (27), it should be said that the $p(i(\tau), j(\tau))$ correspond to the probabilities with the tilde, otherwise the following is not clear.
- Please check the indexes in equations from (28) to (31). It seems that k should be replaced by $k-1$ in several places. Some indexes are written improperly. Moreover, it seems that the factors in the product in Eq. (29) have to be rewritten to effectively represent the ordering in the reverse trajectory (perhaps I am wrong, but check this). I report here below the copied-pasted part of the SM with indicated in yellow the points to be checked/corrected. Please take care to the consistent use of “,” and “;” in the argument of the conditional probabilities; there is a mixed notation here.

From the Markov property, we can rewrite the probability as

$$P((i_k, j_k), t_k; \dots, (i_0, j_0), t_0) = \prod_{l=0}^k p((i_{l+1}, j_{l+1}); t_{l+1} | (i_l, j_l); t_l) p((i_0, j_0); t_0) \quad (28)$$

The probability of discretized time-reversed trajectory can be written as

$$P((i_0, j_0), t_k; \dots, (i_k, j_k), t_0) = \prod_{l=0}^k p((i_l, j_l); t_{l+1} | (i_{l+1}, j_{l+1}); t_l) p((i_k, j_k); t_0) \quad (29)$$

The log-ratio of these probabilities can be written as

$$\log \frac{P((i_k, j_k), t_k; \dots, (i_0, j_0), t_0)}{P((i_0, j_0), t_k; \dots, (i_k, j_k), t_0)} = \ln p((i_0, j_0)) - \ln p((i_k, j_k)) + \sum_{l=0}^k \log \frac{p((i_{l+1}, j_{l+1}); t_{l+1} | (i_l, j_l); t_l)}{p((i_l, j_l); t_{l+1} | (i_{l+1}, j_{l+1}); t_l)} \quad (30)$$

Let us add $\log \prod_{l=0}^k \frac{j_l! c^{j_l}}{j_l! c^{j_l}}$ to the right-hand side of the equation. This term is identical to zero, so the equality does not change. We rewrite the right-hand side as follows

$$\begin{aligned} \log \frac{P((i_k, j_k), t_k; \dots, (i_0, j_0), t_0)}{P((i_0, j_0), t_k; \dots, (i_k, j_k), t_0)} &= \ln p((i_0, j_0)) + \log \left(\frac{j_0!}{c^{j_0}} \right) - \ln p((i_k, j_k)) - \log \left(\frac{j_k!}{c^{j_k}} \right) \\ &+ \sum_{l=0}^k \log \left(\frac{j_l!}{j_{l+1}!} c^{j_{l+1} - j_l} \frac{p((i_{l+1}, j_{l+1}); t_{l+1} | (i_l, j_l); t_l)}{p((i_l, j_l); t_{l+1} | (i_{l+1}, j_{l+1}); t_l)} \right) \end{aligned} \quad (31)$$

- The sentence after Eq. (31), reported here below, is not clear:

By performing the limit $k \rightarrow \infty$ the left-hand side expression becomes the quantity R in the main text. Since

$$p((i_{l+1}, j_{l+1}); t_{l+1} | (i_l, j_l); t_l) / (t_{l+1} - t_l) \rightarrow \tilde{w}_{ii'}^{jj'} \quad (32)$$

and $p((i_0, j_0)) = \tilde{p}_{i_0}^{(j_0)}$, we find that the first term is simply the entropy difference between initial and final state. The second term on the right-hand side becomes the entropy flow along the stochastic trajectory, as defined the above. Therefore, the right-hand side becomes the entropy production difference Δs_i along a stochastic trajectory $(i(\tau), j(\tau))$.

First, I cannot find a quantity ' R ' in the main text. Then, I guess that with 'first term' at the right-hand side it is meant the first four addends as a whole, while 'second term' refers to the summation. By taking into account Eq. (21) of the SM, it seems that the first term relates with to the entropy difference plus the variation of the molecular order (not only to the entropy difference). Due to the improvement of Stirling's approximation, the formula for the stochastic entropy has been changed with respect to the one in the previous version of the manuscript. Now there is an extra term equal to $-(j(\tau)-1)$ in Eq. (21)... For getting the final statement about the whole right-hand side, it seems that the summation in Eq. (21) should be minus the entropy flow (not the entropy flow). Moreover, turning back to the 'probabilities' without tilde, it is not clear how the scaling factors (i.e., the ratios of the kind j/l in the line below Eq. 26) cancel, and if they really do cancel each other. I would ask the Authors to give the explicit passages, so to check the results or make possible corrections. In particular, I strongly suggest to explicitly write Eq. (32) for both the numerator and the denominator so to better follow the final steps and check the results.

In short, please check carefully the whole derivation in the SM from Eq. (27) on.

Minor points:

3) In the SM, but also in the main text, the Authors use the expressions "change of entropy production" (for Δs_i) and "change of entropy flow" (for Δs_e). I feel hard to understand the meaning of the combinations change-production and change-flow (which double the sense of variation). With reference to a finite piece of stochastic system's trajectory, should they simply be "change of entropy" and "entropy flow"?

4) In the SM, regarding Eq. (19) it should be said that this term is zero. It is implicit in the line after the equation, but it is better to state this explicitly.

5) I am wondering if Eq. (25) in the main text is really necessary here. The "time derivative" of the Dirac delta functions is quite unsettling here, without resorting to its elaboration given in the SM. Just think if, for sake of clarity, the text can stand without Eq. (25).

Very minor points:

6) The word "recently" sounds a bit odd in "Recently, a lot of attention has been given to finite-size corrections of chemical potential [37, 38] and non-equilibrium thermodynamics of small chemical networks [23-26]." Most of cited papers date back to 30-40 years ago.

7) Concerning the notation about the particle state, the Authors use both $x^{(j)}$ and $s^{(j)}$... Is this necessary or redundant? To become confident with the notation takes some effort... Any simplification is welcome.

8) After Eq. (4), the normalization condition is given without having defined the 'probabilities' $p_i^{(j)}$:

Normalization is given by $\sum_{ij} j p_i^{(j)} = 1$. Therefore, the quantity $\tilde{p}_i^{(j)} = j p_i^{(j)}$ can be interpreted as the probability

Please first define the $p_i^{(j)}$ (as it was in the previous version of the text).

9) Check the readability flow in some of the new parts. There are also some typos. For instance:

- In the caption to Fig.1. "The inset shows how the ratio (...)". Something is missing in the sentence, of the 'how' has to be removed.
- At the end of page 5, "ordinary fluctuation theorems with where (...)"
- In the SM, the sentence "We denote the number (...)" before Eq. (1) is hardly readable.
- In the SM, "The of the double system (...)" before Eq. (2).
- In the SM, "disrtibution" before Eq. (4).
- In the SM at page 3, "The maximality axiom is actually result of a similar (...)"
- In the SM, many bibliographic references appear with a ?

Response to referee 2

We want to thank referee 2 for his/her helpful comments. Below we reply point by point to concerns raised in his report:

I think there a typo in the normalisation condition left column page 2 before Eq. (1).

We corrected the typo in the normalization condition.

Before the green text page 3, it is said that the results are valid for any (\square). I am not sure what results are referred to there. One of the needed properties of entropy, extensivity, is shown to work only with $\square \propto \square$ in Eq. (6) of the SM. If (\square) can be anything I am not certain that the concept of "entropy per particle" is well defined.

We added some clarification. Extensivity means that if all extensive quantities double, then the entropy also doubles. Since the number of boxes corresponds to the system's volume, it must also double when we double the system. Therefore, the concentration remains constant, and the entropy is extensive. When b does not change linearly with n , we do not consider a scaled version of a system but another system with a different particle concentration. From this point of view, the entropy presented in the paper is always extensive.

Bottom of page 4 it is stated that "we do not restrict ourselves to the case of thermodynamic limit". Here again, I am not sure what is referred to. The Eqs for \square substituted into Eq.(19) do come from derivations relying on the thermodynamic limit. Likewise, Eq. (20) comes from differentiation of Eq. (9) which uses the thermodynamic limit.

We added a discussion to the conclusions. Since the derivation of the first and second law of thermodynamics for the non-equilibrium case does not require any assumption about the system's size, we can conclude that the results are valid also for small systems. As we now mention in the conclusions, the same approach was famously used before for the case of Shannon entropy.

In fact Eq. (9) has been revised in the new version but its consequences for Eq. (20) have not been carried out. So, unless I am missing something, some extra terms should appear in Eq. (20) and maybe elsewhere too.

We thank the referee for pointing this out. Yes, former Eq. (20) was flawed, and we conclude that we obtain the second law of thermodynamics in the usual form. We have carefully checked all the other equations, which should be fine now.

I am a bit puzzled by the interpretation of Eq. (28) and the derivation leading to it. The thermodynamic identities being used as well as the reasoning appear to imply that \square and \square can be considered as independent state variables with respect to which one can express entropy variations. This leads me to interpret the authors' work as showing that the canonical ensemble for n particles forming molecules is equivalent to a sub-ensemble with an additional fixed parameter: the number of

molecules per particle, in the thermodynamic limit. This new ensemble is then associated to a brand new thermodynamic potential being $\Omega - \mu$ which explicitly appears in the exponent in Eq.(28). Whether I am correct or not, I felt that the manuscript was not clear on this matter.

We thank the referee again. In the current version, we carefully do all the calculations and found some issues that should be resolved now. Correcting these issues changed the final form of the fluctuation theorems which we hope you find correct.

The examples provided Fig1 and Fig2 seem very interesting but they appear totally overlooked and are only mentioned in passing. In that respect the presentation of the work and its impact seems fairly unbalanced.

We tried to discuss the examples a bit more, but a part of the discussion was left to the Supplementary material due to limited space. We have added another exciting example of soft-matter self-assembly where our results can also be applied. With the extended applicability of our results, we hope that the manuscript is already suitable for publication in Nature Communications.

Additional Response to Referee 2 with respect to response to Referee 1

(here we only mention points where Referee 2 was not satisfied)

I believe here R#1 was expecting an attempt of the authors to apply the procedures described in Refs [2-5] and see what kind of entropy follows. From their reply and amendments, the authors have not complied with this expectation.

We have added a discussion to the SM, where we explicitly show that the entropy satisfies the group composition law described in Refs. [2-5], but due to the lack of symmetry, it does not belong to the class of Z-entropies considered in [2-5]. Considering the second Shannon-Khinchin axiom (SK2) is sufficient but not necessary to obtain a well-behaved thermodynamic entropy. We confirm this by derivation of the first and the second law of thermodynamics of the case of structure-forming systems.

In our opinion, group composition is an important concept in information theory. A generalization to the case when the entropy is not a symmetric function of the probabilities would be a stimulating step towards thermodynamics of complex systems.

Authors have successfully addressed this comment but only for Eq. (9).

We have now addressed the issue of the extensivity of any concentration, as described above.

Response to Referee 3

We want to thank referee 3 for his/her helpful comments. Below we reply point by point to concerns raised in his report:

In the out-of-equilibrium part, I think that more precise statements are due in some points.

- When they introduce the master equation Eq. (18), I presume that the Authors are already thinking in all generality to a driven out-of-equilibrium system (i.e., such master equation is not restricted to a simple free relaxation with constant rates w_{ij}).

If this is the case, the sentence "Assuming detailed balance such that (...)" should be rephrased by referring to the "underlying distribution" that would be attained by free relaxation at stopped protocol. Something like:

Assuming detailed balance even under out-of-equilibrium, that is [equation], requires that the underlying stationary distribution coincides with the distribution obtained from the maximum entropy principle. From this (...)

Thank you. We have changed the text accordingly.

Before Eq. (22) the Authors write "Let us still consider that the detailed balance is fulfilled even for the time-dependent energies". It seems more appropriate to say "Let us assume that (...)", as above. This is indeed a widely accepted *assumption* in stochastic thermodynamics, more than a physical fact certainly true. Indeed, the validity of this assumption could be even more questionable here, where the transitions occur between very different physical states (the particles states) and not between structurally vicinal microstates as it occurs in the typical applications where the system is a manipulated single molecule... Moreover, I think it would more appropriate to refer to 'microreversibility', rather than to 'detailed balance' (of course the two concepts are interrelated). This would be more adherent to the derivation of the fluctuation theorem given in the SM.

Thank you again. We have updated the text.

In the derivation of the fluctuation theorem given in the SM, there are some issues that have to be considered. In short, please check carefully the whole derivation in the SM from Eq. (27) on.

We have carefully recalculated the whole section on non-equilibrium stochastic thermodynamics. The issues are now hopefully resolved. The improvements also lead to a change in the work-fluctuation theorem, which has a similar form with the original Crooks' fluctuation theorem.

In the SM, but also in the main text, the Authors use the expressions "change of entropy production" (for Δs_i) and "change of entropy flow" (for Δs_e). I feel hard to understand the meaning of the combinations change-production and change-flow (which double the sense of variation). With reference to a finite piece of stochastic system's trajectory, should they simply be "change of entropy" and "entropy flow"?

We have changed the terminology accordingly.

In the SM, regarding Eq. (19) it should be said that this term is zero. It is implicit in the line after the equation, but it is better to state this explicitly.

We have stated the equality explicitly.

I am wondering if Eq. (25) in the main text is really necessary here. The "time derivative" of the Dirac delta functions is quite unsettling here, without resorting to its elaboration given in the SM. Just think if, for sake of clarity, the text can stand without Eq. (25).

We have omitted the equation from the main text.

The word "recently" sounds a bit odd in "Recently, a lot of attention has been given to finite-size corrections of chemical potential [37, 38] and non-equilibrium thermodynamics of small chemical networks [23-26]." Most of cited papers date back to 30-40 years ago.

We have omitted the word "recently". Moreover, we have amended some newer applications for the theory of soft matter and nanomaterials.

Concerning the notation about the particle state, the Authors use both $x_{(j)}$ and $s_{(j)}$... Is this necessary or redundant? To become confident with the notation takes some effort... Any simplification is welcome.

In the current version, we denote a state solely by x .

After Eq. (4), the normalization condition is given without having defined the 'probabilities'.

In the current version, we define the 'probabilities' first. Moreover, we take special care to distinguish between the 'probabilities' and the actual probability distribution that sums up to one.

Check the readability flow in some of the new parts ...

We considered all suggestions by the referee.

REVIEWERS' COMMENTS

Reviewer #2 (Remarks to the Author):

I have read with attention the newest revised version of the manuscript entitled "Thermodynamics of structure-forming systems" by Korbelt et al and submitted for publication in Nature Communications.

I am thankful to the authors for having followed some of my recommendations and consequently greatly improve the clarity and quality of the manuscript in my opinion.

However, I am still confused by some directions chosen for the revision. Here are my reasons:

1. While one important "selling point" and novelty of the manuscript appears to be a non-equilibrium treatment of self-assembly, there is surprisingly no explicit example provided which would be relying on or taking advantage of out-of-equilibrium situations. Instead, the authors have added one somewhat detailed example of equilibrium phase diagram of patchy colloids. This example appears to be well treated and consistent with previous works but in my opinion the authors had provided enough equilibrium examples. It is also unclear in terms of presentation why these equilibrium examples appear after the out-of-equilibrium results.
2. Maybe I am wrong but I am not particularly surprised that Crooks' fluctuation theorem holds in systems which are structure-forming. This theorem's assumptions are usually generic enough that the contrary (i.e. it not being valid) would be the surprising thing to report in my opinion.
3. Related to point 1, would there be any possibility to apply the out-of-equilibrium framework being developed to, say, a temperature gradient for a toy model of self-assembly? For example the magnetic coin model.
4. I feel that the comparison with self-assembly theory is needed and welcome. But then I believe that the presented theory being posterior to the works shown in [43] or [44] for example, the conceptual and technical differences --- and in particular the advantages of the present approach --- need to be emphasised (for example why don't the authors use the approach presented in Eqs. (15) and (16) of [44]?). As far as I see it the section dedicated to the relation to self-assembly theory actually shows that the two approaches are consistent with one-another. This is then confusing to decide what is genuinely novel and worth expedient communication in the proposed work.
5. Minor comment: should there be a half factor in the second equality of eq. (25)?

Because of these points I am sorry to say that I still do not recommend publication of the manuscript in Nature Communication.

My current appreciation is that the current focus and take-home message of the paper still fail to make me appreciate a quality and novelty important enough to warrant publication in Nature Communications. I would therefore recommend again to re-focus the paper's claims and submit to a more specialised journal (finding illustrated in Fig3 for example is nice but would deserve to be explored further for its own sake).

Report on ***Thermodynamics of structure-forming systems*** by J. Korbelt, S. D. Lindner, R. Hanel, and S. Thurner

I think that the authors have considered attentively the comments and the criticisms raised by the reviewers. At the presentation level, the manuscript has been largely improved in many ways (e.g., thanks to the sectioning and the use of calligraphic notation which helps the reader to keep track of the quantities defined per particle). The authors have also made a significant simplification at a certain point of their derivation (by neglecting an immaterial constant term in the entropy per particle after Eq. 4) which leads to simpler equations. The inclusion of the further example on self-assembly of colloidal particles is pertinent and useful; the results in Fig. 2 are clear and interesting.

Concerning my specific criticisms, the authors have responded in appropriate way to all of them. The re-derivation of the fluctuation theorem now leads to a more “natural” solution (Eq. 32) which is in accord with Crooks’ fluctuation theorem with the partial free energy introduced by the authors; in a sense, this is a check of self-consistency.

On this basis I can recommend the publication of this article. There are however very minor points that the authors should consider. Please see my remarks below.

1) At page 1, the authors say that the theoretical framework about nonequilibrium self-assembly is still lacking. This is not correct, and anyway is a drastic statement. This is a growing field in statistical physics, and important results have been already achieved. Here in the introduction (and/or at the end of the Conclusions) the authors should at least mention the recent work of Nguyen and Vaikuntanathan:

Design principles for nonequilibrium self-assembly, *Proc. Natl. Acad. Sci. USA* **113**, 14231–14236 (2016)

For a class of prototype systems, this work deals with compositional fluctuations under an external thermodynamic drive imposed on the system. The self-assembly problem is faced here from a different angle, but since the authors present their own nonequilibrium thermodynamics of structure-forming systems, it is due to mention other already existing research lines in the field.

2) In the Supplementary Material:

- Please check the statement in the sentence after Eq. (25) concerning the ensemble average. Perhaps the dot should be removed.

Thus, we obtain $\dot{s} = \dot{s}_i + \dot{s}_e$. The ensemble second law of thermodynamics can be recovered by multiplying Eq. (23) by $\frac{1}{\Omega}$ and summing over i, j .

- The authors should make some further effort to explain the physical meaning of the quantity in Eq. (26). They call it a “probability” (within quotation marks) of the trajectory $\mathbf{x}(\tau)$. In addition, it should be recalled that when writing Eq. (27), and then Eq. (31), for the reverse protocol it is assumed that microreversibility holds under the external drive.

- In Eq. (34), perhaps Δs_i should be $\Delta\sigma$; just check:

$$\begin{aligned} P(\Delta\sigma) &= \int \mathcal{D}[\mathbf{x}(\tau)] \mathcal{P}(\mathbf{x}(\tau)) \delta\left(\Delta\sigma - \log \frac{\mathcal{P}(\mathbf{x}(\tau))}{\tilde{\mathcal{P}}(\tilde{\mathbf{x}}(\tau))}\right) \\ &= \exp(\Delta s_i) \int \mathcal{D}[\tilde{\mathbf{x}}(\tau)] \tilde{\mathcal{P}}(\tilde{\mathbf{x}}(\tau)) \delta\left(-\Delta\sigma - \log \frac{\tilde{\mathcal{P}}(\tilde{\mathbf{x}}(\tau))}{\mathcal{P}(\mathbf{x}(\tau))}\right) \\ &= \exp(\Delta\sigma) \tilde{P}(-\Delta\sigma). \end{aligned}$$

- After Eq. (37), I think that “By plugging (36) into (38)” should be “By plugging (35) into (33)”.

- In Eq. (30), I think that λ should be replaced by l :

$$\frac{\mathcal{P}(\mathbf{x}(\tau)|j_0)}{\tilde{\mathcal{P}}(\tilde{\mathbf{x}}(\tau)|j_f)} = \exp(\beta w - \beta(\Phi_{j_f}(\lambda(T)) - \Phi_{j_0}(\lambda(0))))$$

- In general, please make a final proof check of this section.

3) There are duplicated references ([52] and [53] are the same).

Response to Referee #2:

1. While one important "selling point" and novelty of the manuscript appears to be a non-equilibrium treatment of self-assembly, there is surprisingly no explicit example provided which would be relying on or taking advantage of out-of-equilibrium situations. Instead, the authors have added one somewhat detailed example of equilibrium phase diagram of patchy colloids. This example appears to be well treated and consistent with previous works but in my opinion the authors had provided enough equilibrium examples. It is also unclear in terms of presentation why these equilibrium examples appear after the out-of-equilibrium results.

According to the advice of the referee, the example section was moved before the non-equilibrium section. The example demonstrating the non-equilibrium properties of structure-forming systems (e.g., experimental verification of the Crooks' fluctuation theorem) is a subject of the ongoing research.

2. Maybe I am wrong but I am not particularly surprised that Crooks' fluctuation theorem holds in systems which are structure-forming. This theorem's assumptions are usually generic enough that the contrary (i.e. it not being valid) would be the surprising thing to report in my opinion.

It is maybe not surprising that the Crooks' fluctuation theorem can be straightforwardly derived for the case of structure-forming systems, but it has not been derived yet. Also note that contrary to the usual Crooks' fluctuation theorem, there is a slight difference. For the case of structure-forming systems, we have to assume that we start from an equilibrium state of a particular cluster size. Consequently, the partial free energy appears in the Crooks' fluctuation theorem and not the total free-energy.

3. Related to point 1, would there be any possibility to apply the out-of-equilibrium framework being developed to, say, a temperature gradient for a toy model of self-assembly? For example the magnetic coin model.

This is the subject of ongoing research. Hopefully, we will be able to demonstrate the non-equilibrium properties of structure-forming systems, such as the magnetic coin model, in one of the upcoming papers.

4. I feel that the comparison with self-assembly theory is needed and welcome. But then I believe that the presented theory being posterior to the works shown in [43] or [44] for example, the conceptual and technical differences --- and in particular the advantages of the present approach --- need to be emphasised (for example why don't the authors use the approach presented in Eqs. (15) and (16) of [44]?). As far as I see it the section dedicated to the relation to self-assembly theory actually shows that the two approaches are consistent with one-another. This is then confusing to decide what is genuinely novel and worth expedient communication in the proposed work.

We have shown that our approach is consistent with the theory of self-assembly. Our approach can be, on the other hand, understood as an extension of the theory of self-assembly. We are able to calculate probability distributions for each unique state of different cluster size, not only the cluster-size distribution. We have also shown that the approach does not have to be used only in the soft-matter self-assembly but also in other structure-forming systems as chemical networks, magnetic gas or spin models with bond states.

5. Minor comment: should there be a half factor in the second equality of eq. (25)?

Eq. (25) is correct, the second term in the second equality comes from the normalization of the transition rates, $\sum_{i=1}^n w_{ij} = 0$, from which we get that

$$w_{ii} = -\sum_{i \neq j} w_{ij}.$$

Response to Referee #3:

- 1) At page 1, the authors say that the theoretical framework about nonequilibrium self-assembly is still lacking. This is not correct, and anyway is a drastic statement. This is a growing field in statistical physics, and important results have been already achieved. Here in the introduction (and/or at the end of the Conclusions) the authors should at least mention the recent work of Nguyen and Vaikuntanathan: Design principles for nonequilibrium self-assembly, Proc. Natl. Acad. Sci. USA 113, 14231-14236 (2016) For a class of prototype systems, this work deals with compositional fluctuations under an external thermodynamic drive imposed on the system. The self-assembly problem is faced here from a different angle, but since the authors present their own nonequilibrium thermodynamics of structure-forming systems, it is due to mention other already exiting research lines in the field.

Thank you for this comment. We changed the sentence and added some relevant references mentioning non-equilibrium self-assembly.

- 2) In the Supplementary Material:

- Please check the statement in the sentence after Eq. (25) concerning the ensemble average. Perhaps the dot should be removed.

Yes, indeed. We removed the dot.

- The authors should make some further effort to explain the physical meaning of the quantity in Eq. (26). They call it a "probability" (within quotation marks) of the trajectory $x(\tau)$. In addition, it should be recalled that when writing Eq. (27), and then Eq. (31), for the reverse protocol it is assumed that microreversibility holds under the external drive.

We added a simple explanation and recalled the assumption of microreversibility.

- In Eq. (34), perhaps Δs_i should be $\Delta \sigma$; just check: - After Eq. (37), I think that "By plugging (36) into (38)" should be "By plugging (35) into (33)".

Thank you. We have changed Δs_i to $\Delta \sigma$.

- In Eq. (30), I think that λ should be replaced by 1:

Lambda was replaced by 1.

- In general, please make a final proof check of this section.

We checked this section carefully once more.

3) There are duplicated references ([52] and [53] are the same)

We updated the references and deleted the duplicates.